# Location of Coworking Spaces (CWSs) Regarding Vicinity, Land Use and Points of Interest (POIs)

**Marco Hölzel \*, Kai-Hendrik Kolsch and Walter Timo de Vries**

Chair of Land Management, School of Engineering and Design, Technical University of Munich (TUM), Arcisstr. 21, 80333 Munich, Germany; kai.kolsch@tum.de (K.-H.K.); wt.de-vries@tum.de (W.T.d.V.)
\*   Correspondence: marco.hoelzel@tum.de; Tel.: +49-89-289-22565

**Abstract:** Background: The place of work is, besides the place of residence, a main travel destination in the course of the day for working people, who make up the majority of western European societies. Other daily destinations, such as those for childcare, social activities, and buying groceries, are spatially related to both of these. This article aims to detect if and how the character of the neighbourhood and the associated land use is related to the location of coworking spaces. Specifically, we investigate the spatial relation between coworking spaces (CWSs) in peripheral and non-peripheral regions to specific points of interest (POIs). These POIs could be daily destinations relevant for a common lifestyle of working people. The data rely on identifying the location of CWSs (peripheral/non-peripheral, land use) in Germany and relating the location of CWSs to the location of POIs using georeferenced data. The results show an accumulation of CWSs and POIs in non-peripheral regions and residential areas and a higher number of specific POIs in their vicinity. From these results, we infer that a relatively higher number of specific POIs in the vicinity of CWSs makes it more likely to use this service and thus provides specific advantages to users of CWSs. If work is performed in a CWS close to the place of residence, other daily destinations could be reached in a short time and the spending capacity could remain in the local economy. The quality of life could increase, and the commute is shrinking with effects on traffic, carbon emission, and work–life balance. Further research could investigate whether this also occurs in an international context, and could focus on developing social-spatial models, by making of use remote sensing. In this way, one could measure the impact on public space and on the neighbourhood of CWSs more quantitatively.

**Keywords:** rural development; depopulation; diversification; sustainable development goals; co-working; points of interest; urban planning; 15-Minute City

## 1. Introduction

Villages and town centres, especially those located in rural, non-metropolitan regions, and, to a lesser extent, outskirts of metropolitan regions, face several problems. The number of inhabitants is shrinking [1,2], causing vacancy of houses and shops [3] and leaving land either un- or underused [4,5]. For those who remain, work opportunities tend to be in towns [6], leading to both monofunctional and structurally weak villages ('villages without people') as well as monofunctional villages in peripheral areas ('sleeping villages') [7,8]. As professional life and private life is therefore disconnected spatially, people have to travel to work, re-enforcing the village decline and increasing commuting behaviour [9]. Despite these trends, many of these commuting village residents still prefer to enjoy the comfort of a private garden [10,11] and are still dreaming of an individual, detached house [11]. The concept of a 'Garden City' by Ebenezer Howard [12,13] (late 19th century) aimed to avoid slums and protect the population from unhealthy environmental conditions (such as polluted air and water, which was often caused by industrial sites in the vicinity of residences). In combination with the ideas of the

'Lebensreform' [14] from the mid-19th century, which aims to bring human life back to nature, this led to a trend of separating the place of work (factories, plants, etc.) from the place of residence. The spatial separation of the workplace and residence place derived several concepts of a modern city, such as Ebenezer Howard's 'Garden City', Tony Garnier's 'Cité Industrielle', Frank Lloyd Wright's 'Broadacre City', and Le Corbusier's 'Ville Radieuse' [15]. However, such 'modern' towns additionally lead to large volumes of daily commuting, traffic jams, additional road constructions, empty and sleeping villages, and gradually to more $CO_2$ emissions [16]. In the last decades, the amount of work performed in factories by blue-collar workers or in agriculture has been shrinking in Western societies [17], and the reason for the separation of the place of work and the place of residence, to protect people from harmful emissions, is no longer necessary to this extent. This can bring jobs, especially in the tertiary sector, closer to the place of residence.

Given the above, the objective of this article is to identify concepts of white-collar office work, which could bring the place of work into the vicinity of the place of residence.

Despite the significance of these concepts at the time and during the 19th and 20th century, for the 21st century, many of these ideas, and subsequent city and village designs have, however, led to multiple problems for villages, such as the vacancy of land and properties, sleeping and mono-functional villages, traffic, land-taking, and environmental problems. One of the contemporary alternatives to combat these problems could be sustainable coworking spaces, located in rural villages and mixed-use areas, with a versatile range. The justification for this option is that it would bring more vitality to the villages and thus enable more economic and social development. The degree to which this assumed effect is valid is, however, so far unknown. Therefore, this article aims at deriving which factors (location, amenities/services in the vicinity, etc.) could contribute to successful or unsuccessful coworking places in the sense of vitality, versatility, and sustainability.

Considering the strong relationship between the place of residence and the place of work on the one hand, and the frequent combination of the commute with other destinations on the other hand, it seems to be relevant where coworking spaces are located and which other potential destinations are located. Assuming that coworking spaces are increasingly spreading not only in large cities but also in rural areas, it seems important that they are not established somewhere, e.g., in an industrial area, but rather where they are easily accessible and can be combined with other destinations on multipurpose trips. Based on these relations, we have formulated the following research questions:

1.  Where are coworking spaces (CWSs) located, in peripheral or non-peripheral regions?
2.  What kind of land use is characterizing the surrounding of coworking spaces?
3.  Which amenities, services and offers (specific POIs) are located surrounding CWSs, and where, that can be relevant for users and tenants?

Amenities**,** services and offers (specific POIs) in the vicinity of CWSs could attract users and tenants to CWSs, and at the same time, the neighbourhood also could become more attractive due to a higher frequency of people, to which the CWS can also contribute.

This article contains the subsequent sections, in line with the research questions. It first describes the (German) legal, social and historical context in which the separation of work and residence developed. The following section (Material and Methods) explains the data collection and data analysis methods to address the main research questions. The next section summarizes the main findings and results, followed by a discussion section which relates the findings of the research to the existing literature. The limitations of the research are validity in the next section, and recommendations for further research are drawn. The final concluding section provides the main answers to the research questions and provides recommendations for the practice.

This article provides an analysis of the spatial relation between coworking spaces (CWSs), land use and POIs, which are relevant for working people for daily use. The

relevance of spatial proximity and the densities of amenities in different land-uses can be seen here. We consider our findings relevant for a spatial planning approach that aims to avoid $CO_2$ emissions and other disadvantages of motorized individual transport (MIT) and to make rural areas and inner-town locations more attractive through offers, services and, e.g., coworking spaces, which are relevant in the daily routine of working people.

Germany was chosen as the region of interest, as data on CWS locations and POIs were available in a sufficient quantity and quality.

## 2. Literature Review of Legal, Social and Historical Context of Separation between Work and Residence

This spatial relation leads, among other things, to dormitory towns [7,18] and an increasing volume of traffic [19]. A spatial redistribution of knowledge work in the tertiary sector, made possible by digitalization [20], could enable a stronger provision of labour in rural areas and villages [21].

### 2.1. Separation between Work and Residence

The spatial relationship between the place of work or education and the place of residence is the most important spatial relationship in people's lives besides the issues of daily needs and social life or leisure activities [22]. There are several explanations for why there is currently a separation between work and residence. Historically, the separation of the workplace from the residence place arose during the era of industrialization [23]. Before industrialization, the place of work was close to or within the place of residence [24–26]. Industrialization's demand for labour attracted people looking for employment, who settled close to the workplace for easy access to life where the work was.

### 2.2. Dwelling and Cities of the Modern

Following the conceptual idea of the functionally separated city [12,27], new housing estates, factories and production plants were built on separate locations, with the aim to protect people from harmful emissions. This concept, however, caused a number of problems for residents and working people [23]. To deal with these problems, several planners with philanthropic aims [28] developed a city design with a functional separation to protect people from environmental harm [15,27]. In Germany, this conceptual separation is legally manifested in the German Building Use Ordinance— "Baunutzungsverordnung (BauNVO)". Legally, the place of living is defined as the place of first or second residence (Federal Registration Act—"Bundesmeldegesetz (BMG)" § 20 habitat, habitual residence or domicile [29]. This is the apartment or house—a home where people sleep regularly and run a household. Contrastingly, the place of work (or the place of employment) is the formally registered location of the employer or a branch of the employer. According to the Trade, Commerce and Industry Regulation Act— "Gewerbeordnung (GewO)" § 106, an employer has the right to determine where an employee is to perform the work [30].

### 2.3. Functionally Separate Areas

In the context of Germany, this idea translated into so-called 'core areas' following the German Building Use Ordinance § 7 core areas (in German: 'Kerngebiete'). Core areas primarily serve to accommodate commercial enterprises but can also define the designation of central economic, administrative and cultural facilities [31]. With the Building Use Ordinance of 1962, dwellings and residential facilities were largely excluded in core areas [32]. As a direct consequence, the number of inhabitants in central parts of cities and towns [33,34] shrunk. Another direct result was the preference for non-residential facilities, as the economic gains were significantly higher for commercial facilities such as offices and retail [35,36].

*2.4. Commuting Is a Consequence of the Separation of Home and Work*

Currently, however, being mobile and travelling between the place of residence and the place of work, by public transport and cars, makes the separation of the place of work and the place of living increasingly possible, yet it also leads to an increase in daily commuters [22,37]. In general, one could state that the importance of the physical distance being a barrier is declining. This process is described as "distance decay" [38]. It enables people to enjoy more greenery and gardens and to build new detached houses at the outskirts of towns and villages.

Socially, the separation of the place of work and the place of living causes the commute, which is sometimes just a short bicycle ride, but often a commute by train or car, to take up to more than one hour, and this is steadily increasing [9]. The number of commuters is also rising [39]. Commuting is reported to lead to unhappiness and stress, especially if the commute is not by active modes of transportation (cycling, walking) [40,41]. People perceive it as a waste of time, and many are feeling guilty for the ecological damage of $CO_2$ emissions caused by commuting [16,42,43]. Performing the commute by car still emits $CO_2$/greenhouse gas [44], demanding parking and road space [45]. This counteracts sustainable transport (as recommended by the United Nations Secretary-General's High-Level Advisory Group [46], Sustainability Strategy of Germany [47], and the National Platform Future of Mobility [48]).

If the work location is close to the residential location, and/or when the commute can be performed on foot or by bicycle, commuters tend to spend money in the vicinity for personal commercial activities, and thus contribute to the small and medium economy within the vicinity of the work place [49]. Commuters can be considered as relevant for the local economy [50–52].

*2.5. Sprawl and the Donut-Effect as a Consequence of Commuting*

The result of a rising amount of commuters is an emerging suburb, suburbia [53] or exurb [54], manifested in urban sprawl. The sprawl not only grows in the immediate vicinity and surrounding of cities and metropolises, by there are also edge cities [55] reaching out in the country and there are likewise rural sprawl enabled mainly by car traffic [56]; inner-villages are decaying [57], and this causes the donut effect [58–61], wherein villages grow with new detached houses with shopping and commercial districts at the outskirts, whilst houses and shops in the previous village centre become abandoned. The central village and town areas decay while the outskirts expand and the built-up town structure forms a donut—the so-called donut-effect [60,61].

*2.6. Knowledge Work and Digitalization*

With digitalization and the rising share of non-physical, knowledge-based work [62] from the so-called 'creative class' (following the terminology of Richard Florida [63]), the place of work is decoupled from the location of the employer, which enables more remote work and telecommuting [64–66]. Besides the employed white-collar worker, there is a rising number of freelancers, which perform knowledge-based work and are already not bound to the location of their clients [67].

The current society also needs highly specialized knowledge workers [68], yet many of them cannot find an appropriate job at their desired location [69]. Traditionally, they would have to relocate to the place of the employer. However, with the opportunities of modern remote work for telecommuting, they could in theory be free to choose the place of residence according to their personal preferences and thus be able to travel to the location of the employer for specific purposes only, such as for in-person meetings [70]. This behaviour was visible during the forced COVID-19 lockdown [71,72], although it also led to social isolation [73] and people being stressed about coalescence of private and professional life or the double task of remote work and home schooling [74].

## 2.7. Coworking Spaces

Performing work in a coworking space could be an alternative option, by separating the place where work is performed from the place where private and family-related tasks occur. A coworking space is a location, similar or comparable to an office, mainly as an open space office, often with a higher quality of design and a more differentiated offer of workplaces, desks, meeting rooms, phone booths, lounges, etc. [75,76], where people are "working alone together" [77] in a social context, with "colleagues" that do not have to have the same employer. As Merkel described it, "Coworking is hence not just about working 'alone together' or 'alongside each other' in a flexible and mostly affordable office space. It is also underpinned by a normative cultural model that promotes five values: community, collaboration, openness, diversity, and sustainability. This 'collaborative approach' is always underlined as a distinctive feature that sets coworking apart from other forms of shared, flexible work setting such as satellite offices, hot desks, coffee shops or business incubators" [78]. However, the boundaries seem to be fluid, and the term "coworking space" is often used by business centres or shared offices alike—or as a specific subtype of business centres [79].

Coworking spaces in rural regions could represent locations to conduct work [21,80]. The attractiveness of coworking spaces not only reflects the attractiveness of the coworking space itself but also reflects the attractiveness of its vicinity [81]. Hence, there is a correlation between job opportunities, depopulation and services offered in a spatial context. The opportunities have already been recognized, and there are some initiatives, such as CoworkLand eG, and programmes that support this. The German funding database [82] identifies around 499 funding programmes under the search terms 'land' and 'digital', one of which explicitly includes the term 'coworking space' [83]. The Federal Ministry of Food and Agriculture supports the idea of rural coworking spaces [84].

Coworking spaces could also be supported by programmes such as LEADER, ZILE 'Integrated rural development in Lower Saxony' [85]. The new coalition agreement of the Federal Government [86] states that "Coworking spaces are a good opportunity for mobile work and strengthening of rural regions". Despite the fact that this policy has not yet seen any concrete activities, at least the intentions for the coming years are clear. At the European level, there are—besides the existent rural development programs (RDPs), such as LEADER etc.—new initiatives such as the 'Long-term vision for the EU's rural areas', the 'Rural Pact', the 'EU Rural Action Plan' [87]; rural coworking spaces are matching with the Priority & Focus Areas 1 and 6 [88]. Several sections of the EU's 'Green Deal' of 'The New European Bauhaus' [89] could support the idea of coworking spaces, especially in rural regions. Last but not least, the European Network for Rural Development (ENRD) provides a 'Rural Coworking Guide' dealing with the general issues in rural regions, business/management models, needs, equipment, networking and communication [90].

There are several websites [91–94] that provide tools to find a coworking space using searching filters with different criteria, including the available equipment, rental price, availability and location. When evaluating the possible locations, it is obvious that most coworking spaces are in urban regions. However, increasingly, there are also coworking spaces in rural areas [80,95–97]. Previous publications also confirm this [97–99], arguing that with the presence of co-workers in rural areas could reactivate the use of previously abandoned houses in rural village centres. Regardless of whether of the location is in rural or non-rural areas, it is still largely unknown to what extent the location of a CWS relates to land use or to the presence of other specific facilities and services. Mariotti et. al. posit that the location of a CWS strongly depends on a particular set of spatial artefacts. Their analysis locates CWSs, regarding the NUTS4 (Nomenclature des Unités territoriales statistiques—since 2005, local administrative units (LAUs) [100]), and found a dominance in urban areas, followed by suburban areas and "…to a lesser extent, peripheral and inner areas"[101]. In addition, specific types of land use may influence the occurrence of CWSs, which has been studied through some research [102–104]. Still, however, these examples are rather isolated and do not reflect a regional or national pattern.

Conducting work in a coworking space is a reflection of the separation of professional from private life [72,105], and an alternative to execute the job from a "Third Place" (other than home or office [76,106,107]. As such, working from a coworking space close to the place of residence could facilitate a better work–life balance whilst avoiding the need to physically commute, thereby creating the possibility to socially isolate [98,99].

CWSs not only offer advantages for their users: the respective nearby areas also benefit from the presence of a CWS, as it brings vitality to the neighbourhood [81,108] and increases spending at local businesses, especially when the trip to and from the CWS is carried out on foot or by bicycle [109,110].

All the aforementioned aspects are geographically related. The distance between the place of residence and place of work (in a coworking space with the provided services) matters. To travel between these geographical destinations in the course of the day is a demanding task. If these daily destinations are located close to each other, the required time and effort is relatively low, which could imply that the distance is more likely to be covered by walking or riding a bicycle [111,112].

*2.8. City Schemes Regarding Vicinity Are Back*

The documented evidence about travel behaviour related to the place of work is, however, fragmented, especially when the workplace is in a coworking space. Additionally, there is still limited evidence about the extent of the spatial inter-connection between the presence of small and medium commercial enterprises in the vicinity of the coworking places. There exist, however, several geographical and planning models that theorize the relationship between work and residence in general. This includes the general planning ideas captured in the Charter of Athens, the neighbourhood idea of Jane Jacobs [53], the models related to points of interest, etc. While the Charter of Athens propagates the separation of functions of the built environment in residential districts, with districts for production and for commerce and leisure, Jane Jacobs follows a different concept wherein people from different backgrounds and origins could meet by reducing boundaries. Points of interest (POIs) provide the potential for people to meet because they are of a more or less common interest. Following this thought, the availability of a high number of POIs close to the place of residence and place of work increases the chance for people to meet other people.

Newly developed areas on the outskirts of towns and villages are separating commercial and residential uses for several reasons. First is the dominating idea of the separation of uses according to the "Garden City" [12], with the ambition to protect people from the harmful emissions of industrial sites. Secondly, the concept of separation became the guiding idea for urban planning in the 20th century and has been incorporated in the "Charter of Athens" (1933) [27], an influential work on planning. The German Federal Land Utilisation Ordinance (Baunutzungsverordnung—BauNVO) still follows the ideas of the Charter of Athens, by defining the specification of land use by allowing only certain listed land uses and prohibiting others that are not listed. This is the legally binding implementation of the goal formulated in the Charter to separate the areas of the city according to their functions [32].

In the time between the concept of separating land uses to protect people from harmful emissions (late 19th and early 20th century) and today, the economy has developed from an industrial to a knowledge-based economy of service and finance [17,113], which has reduced many of the harmful emissions and enabled a borderless use of land where, e.g., commercial and residential uses could directly meet each other and be intertwined with one another [114]. Some current planning schemes, such as the 15-Minute City, are taking this into account, but these concepts are rarely implemented and more traditional functionally separate structures are specified by the legal framework (BauNVO), adopted by municipalities as land use plans and then built. Hereby the land take is mostly above the population growth [115]—if it grows at all [116]. Municipalities, which have planning sovereignty in Germany (German Constitution—GG Art. 28),

finance themselves to a considerable extent through revenue from trade tax (GG Art. 106), which is paid by resident companies. Therefore, municipalities tend to designate large areas for commercial use in order to facilitate the settlement of companies. Partly due to this oversupply, land prices here are often below those for other land uses [117].

The concept of the "15-Minute City" [118] considers locations of immediate daily needs for an individual relevant if they are located within a fifteen-minute time distance. Such points of daily needs and services can be considered POIs [119]. However, daily needs and services highly vary depending on the household and family situation, and it is this complex set of possibilities where various types of destination (grocery store, school, childcare, office (or coworking space) business trips, sport, leisure, recreation, etc.) need to be combined. This creates a city concept in which variety and complexity play a crucial role in constructing space, which is conceptually the opposite of other city construction concepts such as the "Charter from Athens" and a resurgence of the neighbourhood idea of Jane Jacobs [53]. The younger opposites to the concepts of the modern Charter of Athens—with a functional and spatial separation—advocate for the necessity of having close spatial relations, mixed uses and walkability to needs and services in order to foster sustainability, vitality and liveliness. The 15-Minute City may even contain neighbourhoods or communities in which everything is accessible within 5 min [120]. The concept of mixed use and accessibility by vicinity is to be found in the "New Leipzig Charter" as well [121]. For the particular household set of the family and elderly people, such a closer concept of a neighbourhood with walkable distances to all possible services (including health facilities) would be a preferable solution to city designs in which such services are centralised in specific large-scale, high-volume locations [122–124].

Ridwan and Dimas evaluate to what extent land use und local features in the city of Bandung have an effect its the attractiveness to creative people. It was found that proximity to, for example, coffee shops, bars and sport facilities is of significant importance for the attractiveness of higher educational facilities (such as universities or research centres) [125].

Services that are of relevance for daily needs include grocery stores, supermarkets, restaurants, cafés, public transport, bakeries, kindergartens and cinemas, amongst others. These are all places or points—in a spatial sense—and thus have a specific location, reflected as points of interest (POIs) [126]. The above-mentioned POIs (grocery stores, supermarket, etc.) can be combined with trips for different purposes [127,128].

In light of the above-mentioned aspects, we can consider the land use surrounding coworking spaces and the specific networks of services functional if they combine multiple purposes and if they are spatially related to coworking spaces.

## 3. Methods and Materials

There is a wide and rapidly growing range of literature on the subject of coworking spaces, fablabs, etc., as it could be found, e.g., at the Coworking Library [129]; however, we found a limited amount of literature on our research focus—the spatial relation of CWSs and POIs. The theoretical concepts, ideas and models insufficiently capture current realities of remote work and coworking. Additionally, they do not capture the reasoning and justification for certain choices of coworkers. For this reason, this research aims to collect more data on these issues and try to find alternative interpretations. We decided to investigate the research questions by analysing the location of CWSs on different spatial scales: firstly, the general location—peripheral or non-peripheral; secondly, the dominating land use, where CWSs are located; and thirdly, where services and offers, which could be relevant for users of CWSs, are located in the vicinity of CWSs. This investigation should be based on data on the location of CWSs, on land use and on the location of other relevant offers and services. The location of offers and services could be identified by using the available data of POIs, which are partly relevant offers and services.

The use of POIs is, however, useful in the context of this work, because POIs are spatial locations which are relevant, i.e., of interest, to people. POIs are providing a location of a service, an offer, of something else what people could make use of or interact with [130]. In addition to the benefits for users using POIs also provides the opportunity to investigate the degree to which the presence of several CWSs in a specific neighbourhood provides spatial benefits.

POIs are collected by different services, e.g., Google Maps, Foursquare, OSM, etc., with a different number of categories and focus areas. This data collection relied firstly on identifying the locations of existing coworking spaces. Information on the location, name and address of the respective websites of coworking spaces was collected from the website www.coworkingmap.de (accessed on: 08 April 2021), which is a current and comprehensive collection of coworking spaces, with geo-referencing, accessed in early 2021, as a basis of this research. From the source www.coworkingmap.de is a current, comprehensive and reliable source of coworking spaces and mainly focus on coworking spaces in Germany.

The next step of the data collection concerned the classification of coworking spaces. This classification followed both the spatial and non-spatial aspects. At first, coworking spaces were separated into two groups: peripheral and non-peripheral. The definition of peripheral and non-peripheral is based on the harmonised definition of functional urban areas (FUAs) developed by the Organization for Economic Cooperation and Development (OECD) in cooperation with the EU. This definition includes cities and their commuting zones [131]. The OECD defines cities as "a group of local administrative units (i.e., LAU for European countries, such as municipality, local authorities, etc.) where at least 50% of its population live in an urban centre. An urban centre is defined as a cluster of contiguous grid cells of one square kilometer with a density of at least 1,500 inhabitants per square kilometer and a population of at least 50,000 inhabitants overall." [131] According to the definition of Workgroup 1 of the COST Action 'The geography of New Working Spaces and Impact on the Periphery' (CA18214), we decided to classify FAUs below 200,000 inhabitants as peripheral and above 200,000 inhabitants as non-peripheral. In total, there were 96 such FAUs in the dataset of the OECD.

The location of a coworking space is thus classified as "peripheral" if: (a) 1, the coworking space is located outside a metropolitan region, or if they are located within the metropolitan region and this region has less than 200,000 inhabitants; (b) 0, the spaces are located within a metropolitan region with a population more than 200,000 inhabitants. To illustrate these criteria, we list some examples here:

- CWS location outside a metropolitan region—attribute = 1
- CWS location within a metropolitan region that has less than 200,000 inhabitants—attribute = 1
- CWS location within a metropolitan region that has more than 200,000 inhabitants—attribute = 0

To identify the character of the surrounding location of the coworking spaces, we chose the land use database of www.geofabrik.de (accessed on: 08 April 2021), which is based on the OSM database, with the following categories of land use (Table 1). The origin database from OSM is rated as very accurate [132]; these are provided by Geofabrik.de, which are used in other research projects on accuracy [133,134]. Geofabrik.de transferred the OSM database into shapefiles to make the data useable for GIS [135]. In a first step, we joined the location of the coworking spaces with the categories of land use in ArcGIS. For the 80 remaining unclassified coworking spaces, a corresponding OSM class was added by hand using aerial photographs and an existing open-source land use dataset (OSMlanduse.org). This leaves 10 spaces that cannot be clearly classified. Classification of the land use/landcover in the OSM dataset compared with the ATKIS (Authoritative Topographic–Cartographic Information System) shows a high level of completeness and correctness, especially in more urbanized areas [136].

**Table 1.** List of the types of land use from the database www.osmlanduse.org (accessed on: 08 April 2021).

| Categories of Land Use |
|:---:|
| residential |
| commercial |
| industrial |
| retail |
| grass |
| farmyard |
| meadow |
| forest |

Different sources generated the georeferenced information of POIs. One of easiest and most user-friendly ways is to rely on the technical facilities of Google Maps. Additionally, data were collected from German spatial data agencies, such as the BKG—Bundesanstalt für Kartographie und Geodäsie (Federal Agency for Cartography and Geodesy), which provides a specific range of POIs. Furthermore, we relied on open-source services such as *OSM*, which provides a huge amount of georeferenced data by free access. More than 8 million users provide 8.9 billion GPS points, complemented with tags (attributes) [137]. There are no pre-defined categories for POIs in the OSM database, so there could be countless variations of the same kind of POI, with different names, but there is a critical community that takes care of ensuring accurate data, which is entered into the database by registered users. Each entry in the OSM database is stored with the database of the entry. This provides a highly transparent dataset and with that a source of quality assessments [138]. In particular, the accuracy of the shop location, which is an important interaction node for the users of coworking spaces and generates vividness in public spaces, is assessed as "high estimated completeness level of retail stores" [139,140].

Based on the above-mentioned publications and insights, we decided to use the database of OSM processed by Geofabrik.de as the source for the location of POIs. We used the OSM dataset of POIs from Geofabrik.de because they show a lower lack of ambiguity in the classification of POIs. From the OSM database, we selected the POIs with tags, as listed in Table 2 below.

From the OSM database, 2,668,989 POIs of the classes 'pois_free' (not further defined) and 'transport' from all 16 federal states of Germany were loaded in the GIS system. After filtering out the attributes listed in Table 2 ('pois_free' and 'transport'), 742,067 POIs remained. Of these 742,067 POIs, 41,155 duplicates were filtered out. The 500 m radius was chosen regarding the accessibility, walkable distances [122] and the concept of neighbourhood [120]. The spatial join of the 500 m radius around the CWSs reduced the number of POIs to 41,166 POIs as a total set in the radius of 500 m around the coworking spaces. In a normal working day, the primary journeys are to and from the place of work, supplemented by journeys to shops, eateries, pharmacies, local transport facilities, childcare facilities, sports and cultural facilities or similar places [141,142].

These are exported as a shape file. A *1:n* left inner join (spatial join via the geometric relationship "intersect") results in an assignment of the POIs to the respective coworking spaces based on a spatial join used with 500 m circular zones around the coworking spaces. This results in a table with 56,422 entries. The absolute number of POIs is lower because some POIs are located in the vicinity of several CWSs. Twelve coworking spaces do not have a POI in their vicinity and were therefore excluded from the calculation. This results in a total of 6096 entries in peripheral areas and a total of 50,326 entries in non-peripheral areas.

As described above, we identified more than 41,000 POIs in the vicinity of coworking spaces (radius 500 m), excluding POIs located within a 500 m radius of several CWSs. The

listed POIs (Table 2) were joined with the land use categories (Table 1) in a spatial intersection process of the GIS Software.

We have selected the POIs in Table 2 with regard to their usefulness in the everyday life of the working population, 26 out of 135 in the category 'Points of Interest' and 5 out of 10 in the category 'Points of Transport'.

**Table 2.** List of the chosen tags from the processed points of the OSM databased in the categories "Points of Transport" and "Points of Interest".

| From the 10 Tags in the Category "Points of Transport" | From the 135 Tags in the Category "Points of Interest" |
|---|---|
| | bakery |
| | bank |
| | bar |
| | beverages |
| | bicycle_rental |
| | bicycle_shop |
| | biergarten |
| | bookshop |
| | butcher |
| | café |
| | car_sharing |
| | cinema |
| bus_station | community_center |
| bus_stop | convenience |
| railway_halt | doctors |
| railway_station | fast_food |
| tram_stop | greengrocer |
| | kindergarten |
| | kiosk |
| | laundry |
| | library |
| | pub |
| | restaurant |
| | school |
| | supermarket |
| | theatre |

The chosen POI seems to be relevant for a regular interaction with repetitive work at a coworking space.

## 4. Results

### 4.1. Analysis of Peripheral and Non-Peripheral Locations

We first intersected the location of coworking spaces with the shape files of the FAUs provided by the OECD using ArcGIS. We found 149 coworking spaces in peripheral locations, outside of FUAs, according to the definition of the OECD and 562 coworking spaces in non-peripheral locations within FUAs. A majority of 79% of the analysed coworking spaces are located in FUAs or in peripheral regions and 21% outside of FUAs or in peripheral regions within FUAs. Cities and their interconnected region are still the major home for coworking spaces (Figure 1).

Reasons for the dominance of non-peripheral location of coworking spaces are probably the higher population density and the fact that such facilities are used by a rather young, urban clientele.

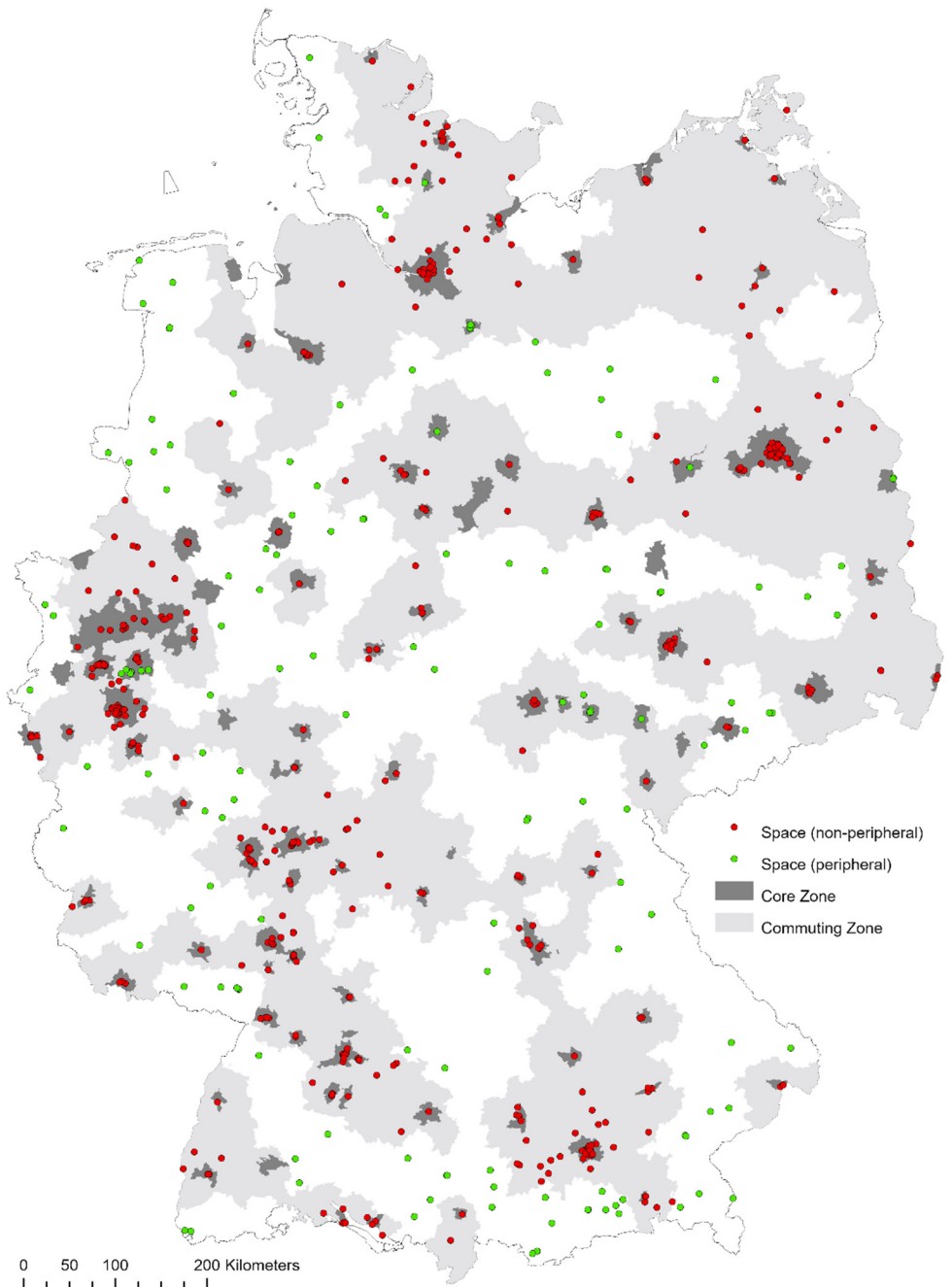

**Figure 1.** Location of coworking spaces (green dots: in peripheral areas, red dots: in non-peripheral areas, dark grey: core zones, light grey: commuting zones) in Germany interlaced with the FAUs defined by the OECD, source: own illustration based on © GeoBasis-DE/BKG (2020), www.coworkingmap.de (accessed on:04. June 2021).

By comparing the location of coworking spaces with the categories of land use (OSM data) via the GIS system, we could classify 701 coworking spaces. Some that did not join the shape files of the land-use categories were classified by analysing areal images from ArcGIS Pro, Google Maps, Google Earth, and www.geofabrik.de (accessed on: 08 April 2021).

*4.2. Analysing the Location of CWS by Land Use*

We found that a majority of coworking spaces is located in residential areas, 63% (450 of 711), 20% in commercial areas, 8% in industrial areas, 7% in retail areas and in sum 2% in more agricultural surroundings such as grass, farmyards, meadows and forests.

Reasons for the dominance of coworking spaces in primer residential neighbourhoods are probably the higher population density and the easier accessibility due to a shorter distance from the place of residence.

Regardless of whether a site is located in a rural or urban area, the analysis shows that a majority of coworking spaces are located in residential areas.

To observe the difference between peripheral and non-peripheral areas in the type of surrounding land use, we separate the examination group into non-peripheral (Figure 2) and peripheral (Figure 3).

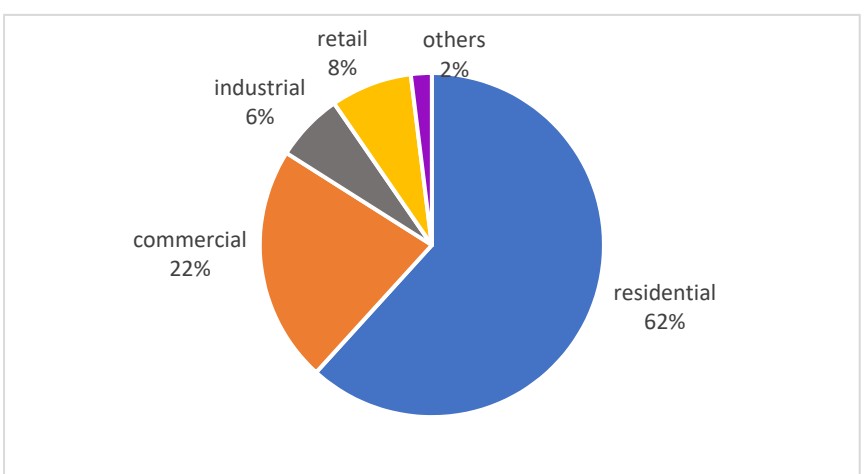

**Figure 2.** Distribution of coworking spaces related to the type of land use for non-peripheral areas.

The majority of land-use types where we could find coworking spaces is residential in non-peripheral areas, with 62% (Figure 2). A total of 22% of the coworking spaces are located in commercial surroundings, 8% in retail-dominated surroundings and 6% in industrial surroundings.

If we take a look at peripheral areas, the picture is changing. Here, we still have the highest share of coworking spaces in surroundings categorized as residential with 69%. A share of 11% of the coworking spaces are located in commercial surroundings; 4% in retail surroundings; 4% in others, such as meadows, farmyards, forest, etc.; and 12% in industrial areas (Figure 3).

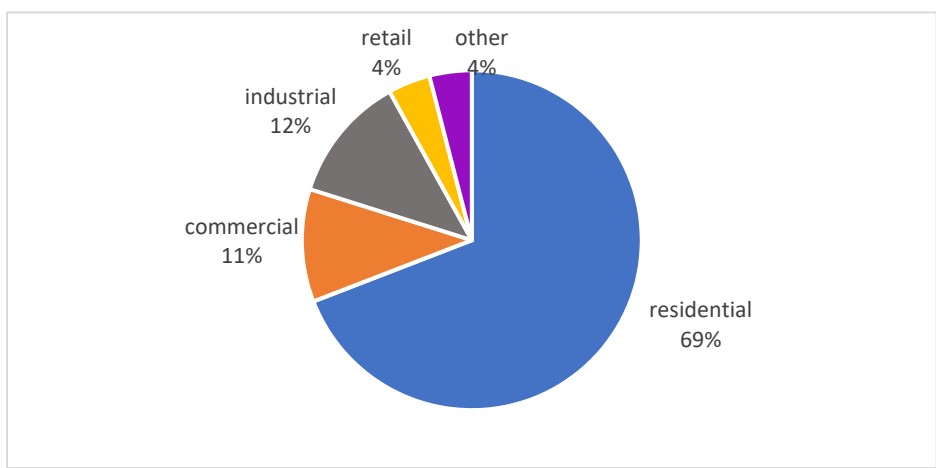

**Figure 3.** Distribution of coworking spaces related to the type of land use for peripheral areas.

Reasons for the dominance of coworking spaces in prime residential neighbourhoods are probably the higher population density and the easier accessibility due to a shorter distance from the place of residence. The lower share of coworking spaces in industrial neighbourhoods could be due to the fact that car use is dominant in rural and peripheral regions and there are more parking spaces available due to the lower density of buildings in industrial neighbourhoods. The halved value for commercial and retail locations of coworking spaces was the same between non-peripheral and peripheral regions. This could be due to the fact that explicit retail and commercial areas do not exist or can be identified less frequently here. The variation in the share of land use (absolute numbers) where CWSs are located is compared in Figure 4, between peripheral, non-peripheral, and in total.

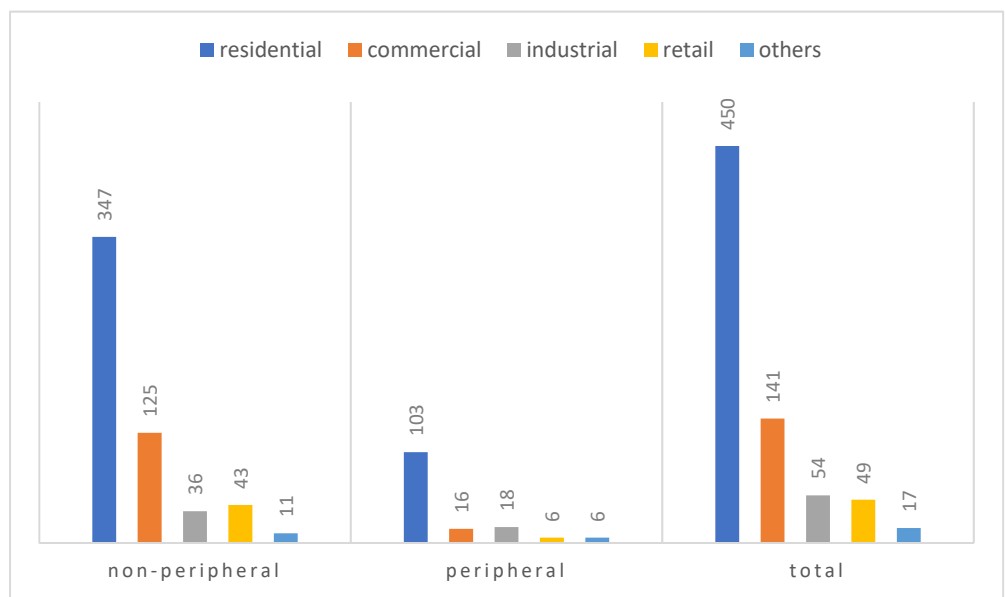

**Figure 4.** Comparison of the distribution of coworking spaces related to the type of land use for peripheral and non-peripheral areas, and in total.

### 4.3. Analysing the Spatial Relation of CWSs and POIs

The POIs listed in Table 2 are relevant for daily needs, regular uses and social or cultural issues. The spatial relation to the place of residence and the place of work is of significance. The POIs from Table 2 and the place of residence and place of work are the main destinations of everyday mobility [51,52]. In our study, the place of work is a coworking space.

A distance of 500–1000 m can be considered a walkable distance [124]. Therefore, we chose a lower limit of 500 m as a walkable distance to ensure the comfort of walking accessibility. By creating a 500 m radius around the individual coworking spaces, we selected the POIs inside this circle as easily accessible and analysed their amount per type.

In the vicinity of an average coworking space, we found more than 14 'restaurant' POIs, as shown in Figure 4, more than 8 'bus_stop', 7 'café', nearly 4 'bakery', and 2,1 'supermarket' POIs, as shown in Figure 5.

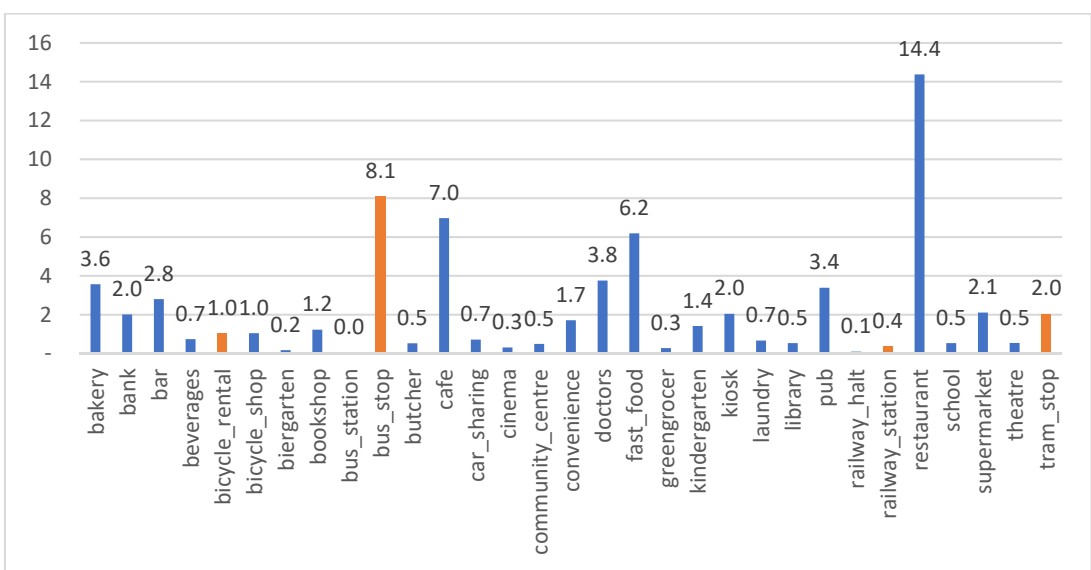

**Figure 5.** Average number of different POIs in the vicinity of 500 m in the surrounding of coworking spaces in peripheral and non-peripheral areas.

The reasons for these findings could be the relatively high number of POIs, such as 'restaurant', 'bus_stop', 'café', etc. in dense and more residential or commercial areas where coworking spaces are mainly located. In further steps, we compared the number of POIs around coworking spaces (radius 500 m) in different locations by their land-use category and regional character as peripheral or non-peripheral.

As visible in Figure 6, all POIs are most available in areas with a dominant land use of 'retail', especially 'restaurant' (27,7), 'fast_food', 'café' and 'bus_stop', with more than 10 POIs in the vicinity. Compared with locations dominated by 'residential' land use, the number of POIs 'restaurant' is below 20, in locations dominated by 'commercial' land use, the number of POIs 'restaurant' is below 10 and in locations dominated by 'industrial' land use the number of POIs 'restaurant' is at 1.6.

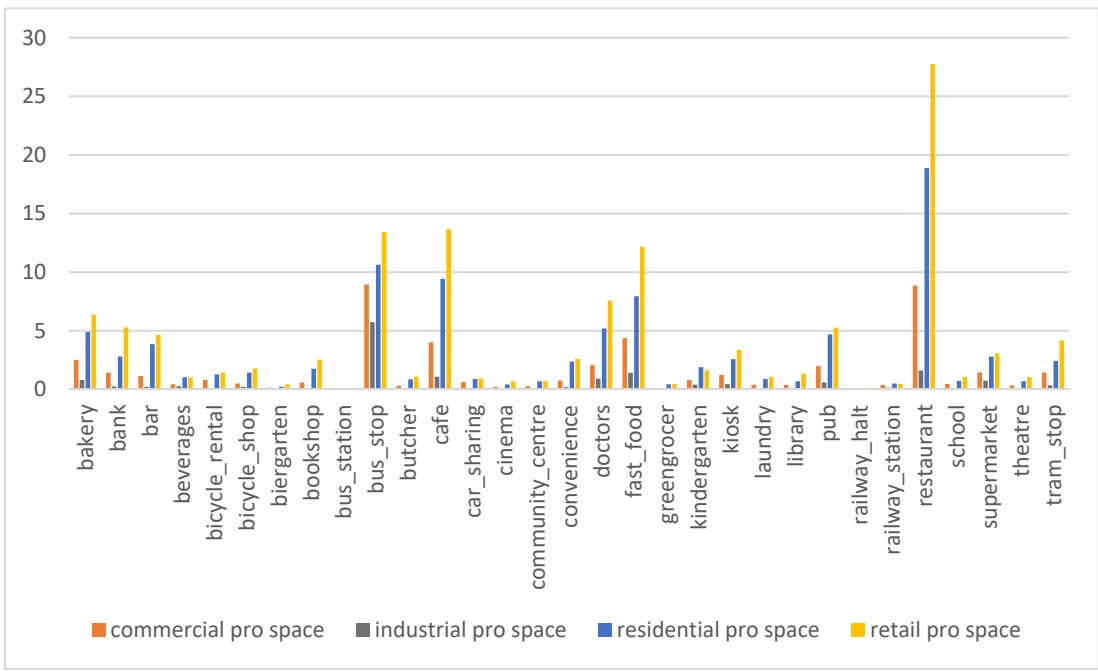

**Figure 6.** Average number of different POIs per coworking space in the vicinity of 500 m in the surrounding of coworking spaces in peripheral and non-peripheral areas compared with the category of land use where the coworking space is located.

For the 'bus_stop' POIs, the dwindling is not very dramatic. The number of the POIs shrinks from 13.4 in 'retail'-dominated locations, to 10.6 in 'residential', to 8.9 in 'commercial' and 5.7 in 'industrial'. POIs attributed to 'pub' are available in areas dominated by 'retail' and 'residential' nearly in the same amount (5.2 in 'retail' and 4.7 in 'residential'), but rarely in 'commercial' areas (2.0) and scarce in 'industrial' areas (0.6).

The regarded POIs seem to be dominant in areas with a high density of populations, such as residential areas or areas with a high number of people visiting, such as retail or commercial areas. That seems to be reasonable because these kinds of POIs need a large number of visitors and customers in order to be economically viable.

In Figure 7 (non-peripheral), all POIs are similar to Figure 6. The most available POIs in areas with a dominant land-use 'retail' are 'restaurant' (28.9), 'fast_food' (12.6), 'café' (13.5) and 'bus_stop' (13.2) in the vicinity. Compared with locations dominated by 'residential' land use, the number of POIs 'restaurant' is 22.2; in locations dominated by 'commercial' land use, the number of 'restaurant' POIs is 9.7; and in locations dominated by 'industrial' land use, the number of 'restaurant' POIs is only at 2.1.

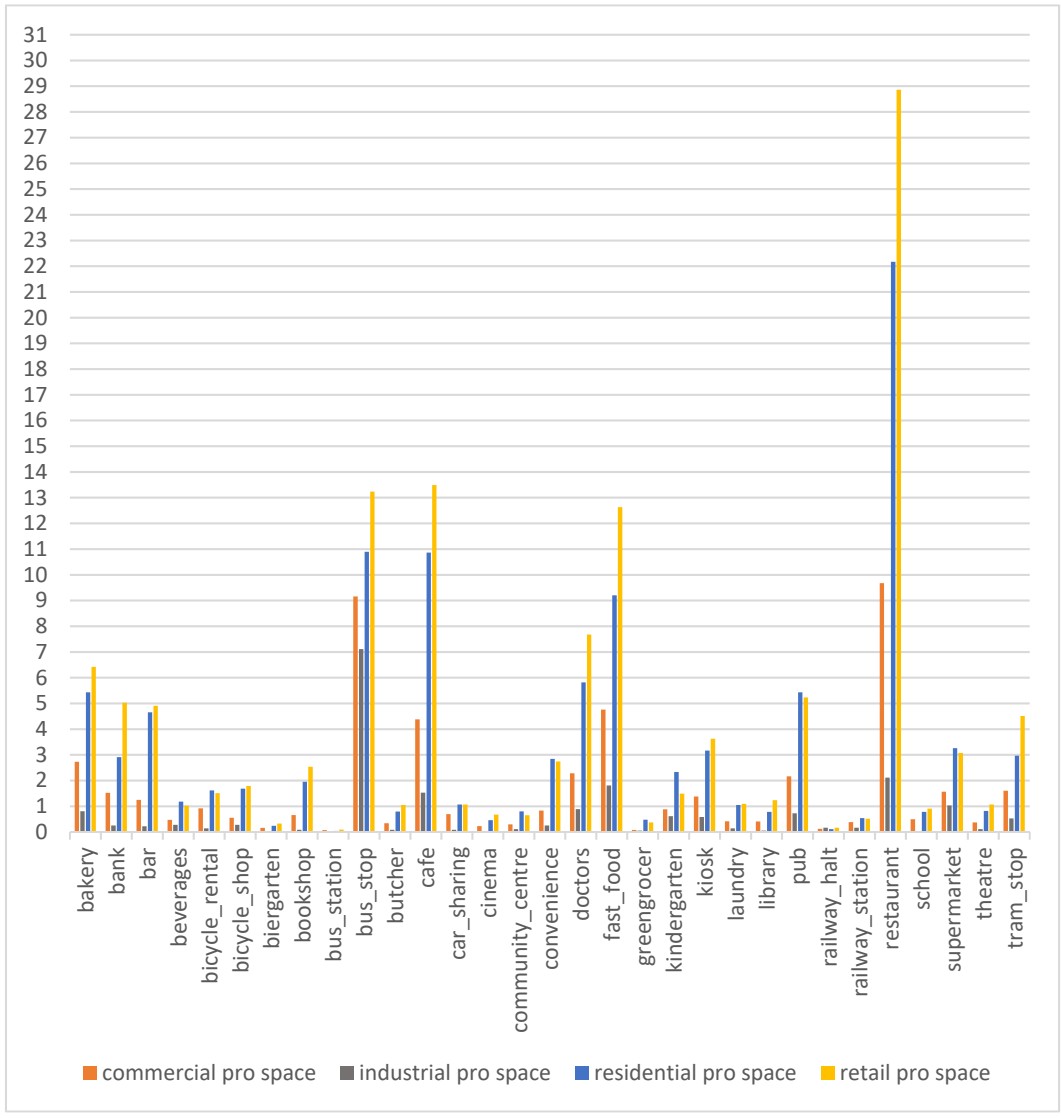

**Figure 7.** Average number of different POIs per coworking space in the vicinity of 500 m in the surrounding of coworking spaces in non-peripheral areas compared with the category of land use where the coworking space is located.

In Figure 8 (peripheral), all POIs are generally similarly distributed to Figure 7. The most available POIs in areas with a dominant land use 'retail' are 'restaurant' (19.8), 'fast_food' (8.8), 'café' (14.8) and 'bus_stop' (14.7) in the vicinity of CWSs (500 m). While the number of 'restaurant' and 'fast_food' POIs is significantly below average, in non-peripheral areas, the numbers of 'café' and 'bus_stop' POIs are above average in non-peripheral areas. Compared with locations dominated by 'residential' land use, the number of 'restaurant' POIs is 7.8; in locations dominated by 'commercial' land use, the number of 'restaurant' POIs is 2.5; and in locations dominated by 'industrial' land use, the number of 'restaurant' POIs is only 0.6.

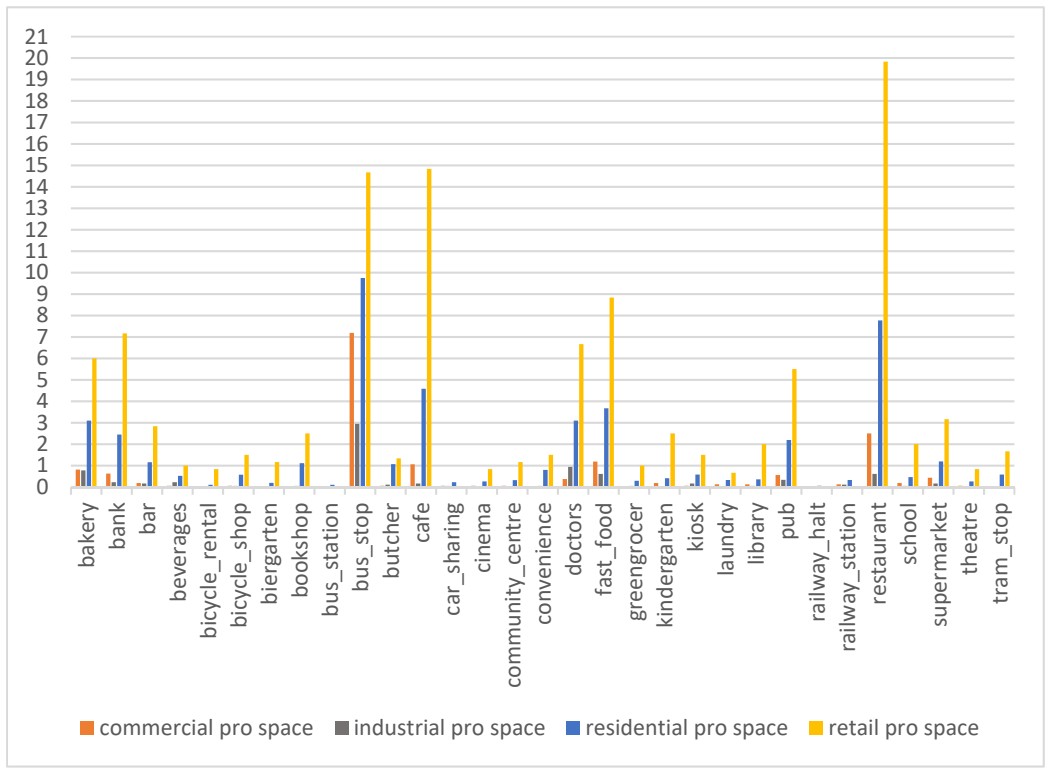

**Figure 8.** Average number of different POIs per coworking space in the vicinity of 500 m in the surrounding of coworking spaces in peripheral areas compared with the category of land use where the coworking space is located.

It seems that the that the POIs are distributed similarly, but with a more extreme distribution. This is probably due to the lower population density and more intensive car use in peripheral, rural regions, which makes it easier to travel longer distances between different functions in different areas.

To have a clear numerical comparison of the number of POIs which could be found in the vicinity of coworking spaces, we create the following table (Table 3), where small numbers, e.g., for bookshops in commercial and industrial areas, could also be recognized.

**Table 3.** List of the average quantity of POIs by land-use type related to peripheral and non-peripheral regions.

| | Peripheral | | | | Non-Peripheral | | | |
|---|---|---|---|---|---|---|---|---|
| | commercial/ space | industrial/space | residential/space | retail/space | commercial/ space | industrial/space | residential/space | retail/space |
| bakery | 0.81 | 0.78 | 3.10 | 6.00 | 2.73 | 0.81 | 5.43 | 6.42 |
| bank | 0.63 | 0.22 | 2.45 | 7.17 | 1.52 | 0.25 | 2.91 | 5.02 |
| bar | 0.19 | 0.17 | 1.16 | 2.83 | 1.25 | 0.22 | 4.65 | 4.91 |
| beverages | 0.06 | 0.22 | 0.52 | 1.00 | 0.47 | 0.28 | 1.18 | 1.02 |
| bicycle_rental | 0.00 | 0.00 | 0.11 | 0.83 | 0.92 | 0.14 | 1.61 | 1.51 |
| bicycle_shop | 0.06 | 0.06 | 0.57 | 1.50 | 0.55 | 0.28 | 1.69 | 1.79 |
| biergarten | 0.00 | 0.00 | 0.19 | 1.17 | 0.16 | 0.03 | 0.24 | 0.33 |
| bookshop | 0.00 | 0.06 | 1.12 | 2.50 | 0.66 | 0.08 | 1.95 | 2.53 |
| bus_station | 0.00 | 0.00 | 0.11 | 0.00 | 0.07 | 0.00 | 0.04 | 0.09 |
| bus_stop | 7.19 | 2.94 | 9.75 | 14.67 | 9.16 | 7.11 | 10.89 | 1.23 |
| butcher | 0.06 | 0.11 | 1.07 | 1.33 | 0.34 | 0.08 | 0.80 | 1.05 |
| café | 1.06 | 0.17 | 4.58 | 14.83 | 4.38 | 1.53 | 10.86 | 13.49 |
| car_sharing | 0.06 | 0.06 | 0.22 | 0.00 | 0.70 | 0.08 | 1.07 | 1.07 |
| cinema | 0.06 | 0.00 | 0.26 | 0.83 | 0.22 | 0.03 | 0.46 | 0.67 |
| community_centre | 0.06 | 0.00 | 0.32 | 1.17 | 0.30 | 0.11 | 0.80 | 0.65 |
| convenience | 0.00 | 0.06 | 0.80 | 1.50 | 0.83 | 0.25 | 2.84 | 2.74 |
| doctors | 0.38 | 0.94 | 3.10 | 6.67 | 2.28 | 0.89 | 5.82 | 7.67 |
| fast_food | 1.19 | 0.61 | 3.67 | 8.83 | 4.76 | 1.81 | 9.20 | 12.63 |
| greengrocer | 0.00 | 0.06 | 0.29 | 1.00 | 0.08 | 0.06 | 0.48 | 0.37 |
| kindergarten | 0.19 | 0.00 | 0.41 | 2.50 | 0.88 | 0.61 | 2.33 | 1.49 |
| kiosk | 0.06 | 0.17 | 0.58 | 1.50 | 1.38 | 0.58 | 3.16 | 3.63 |
| laundry | 0.13 | 0.00 | 0.33 | 0.67 | 0.42 | 0.14 | 1.05 | 1.09 |
| library | 0.13 | 0.06 | 0.36 | 2.00 | 0.41 | 0.06 | 0.78 | 1.23 |
| pub | 0.56 | 0.33 | 2.19 | 5.50 | 2.16 | 0.72 | 5.43 | 5.23 |
| railway_halt | 0.00 | 0.00 | 0.06 | 0.00 | 0.12 | 0.17 | 0.11 | 0.16 |
| railway_station | 0.13 | 0.11 | 0.33 | 0.00 | 0.38 | 0.17 | 0.54 | 0.51 |
| restaurant | 2.50 | 0.61 | 7.77 | 19.83 | 9.67 | 2.11 | 22.17 | 28.86 |
| school | 0.19 | 0.06 | 0.47 | 2.00 | 0.50 | 0.03 | 0.78 | 0.91 |
| supermarket | 0.44 | 0.17 | 1.19 | 3.17 | 1.57 | 1.03 | 3.26 | 3.07 |
| theatre | 0.06 | 0.00 | 0.26 | 0.83 | 0.37 | 0.11 | 0.82 | 1.07 |
| tram_stop | 0.00 | 0.00 | 0.58 | 1.67 | 1.60 | 0.53 | 2.97 | 4.51 |

*4.4. Mapping of CWS Locations and Land Use*

The maps in Figures 9 and 10 (Munich), Figures 11 and 12 (Pfaffenhofen an der Ilm) illustrate the spreading and accumulation of POIs related to CWSs. These examples illustrate several aspects of our rather statistical investigation. On the one hand, it is apparent that CWS sites in large cities have a larger number of POIs in their surroundings, Figures 9 and 10 (Munich), and on the other hand, that land use also has an influence on the number of POIs—there are hardly any POIs in industrial areas, few in commercial areas, and many in residential areas (Figures 9–12).

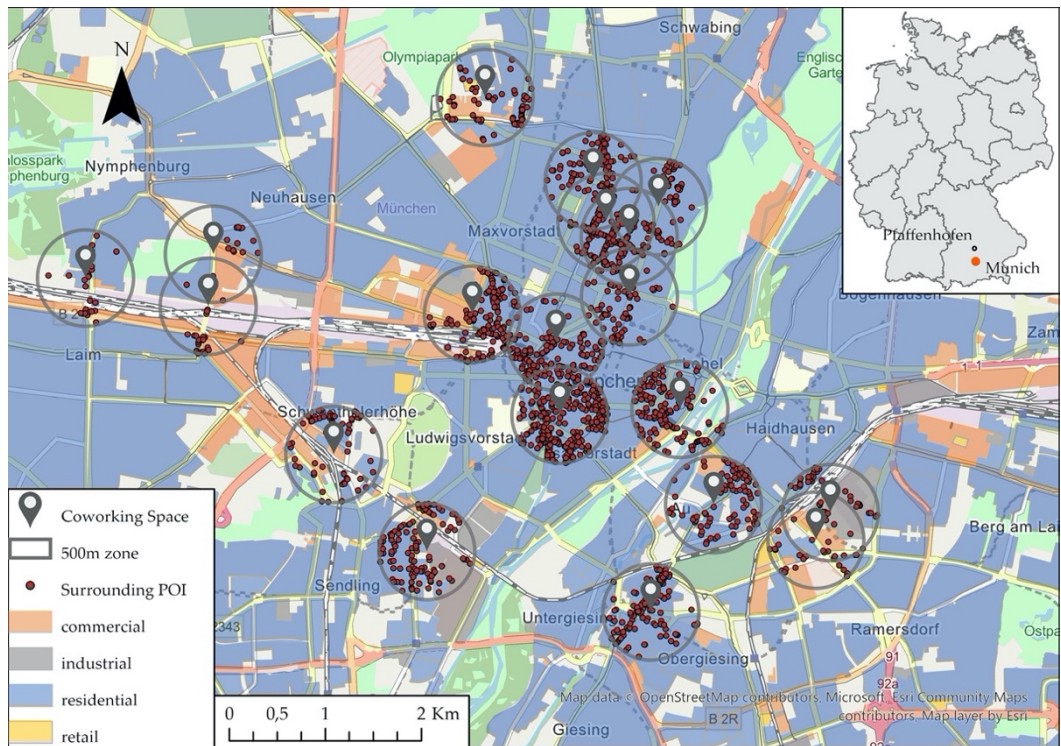

**Figure 9.** Map of POIs as listed in Table 2 in a radius of 500 m around coworking spaces (CWSs) in Munich (without scale), source: ArcGIS® software by Esri, OSM, gefabrik.de, coworkingmap.de, © GeoBasis-DE/BKG (2020).

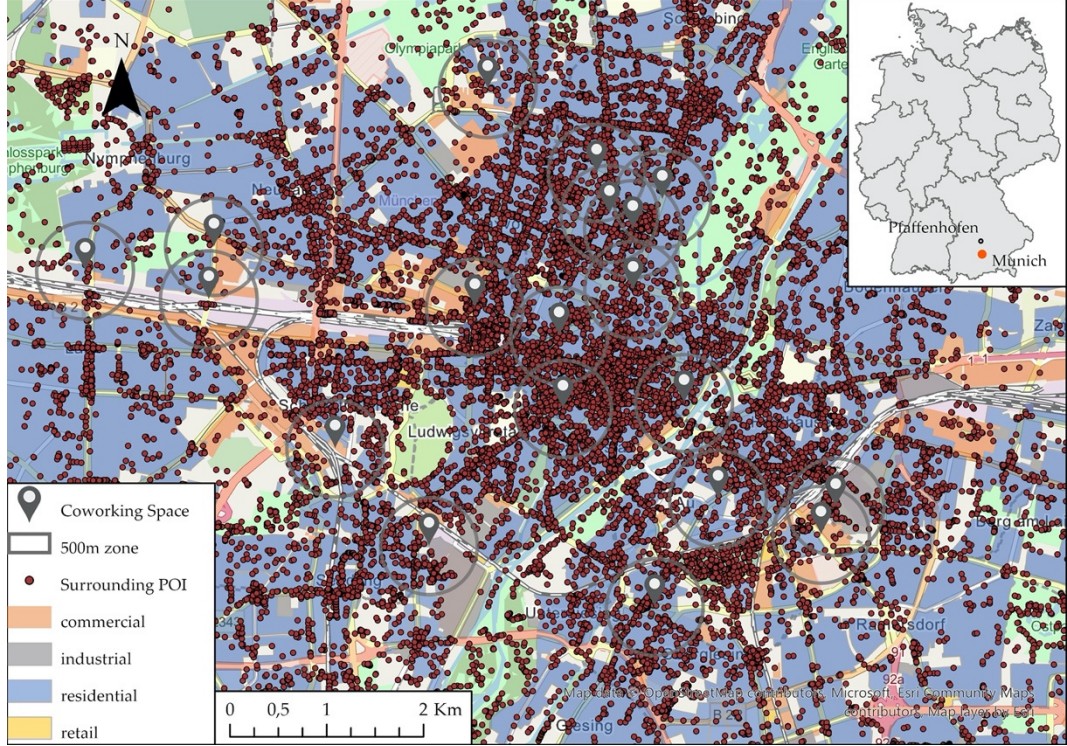

**Figure 10.** Map of all POIs of the category 'pois_free' in Munich (without scale), source: ArcGIS® software by Esri, OSM, gefabrik.de, coworkingmap.de, © GeoBasis-DE/BKG (2020).

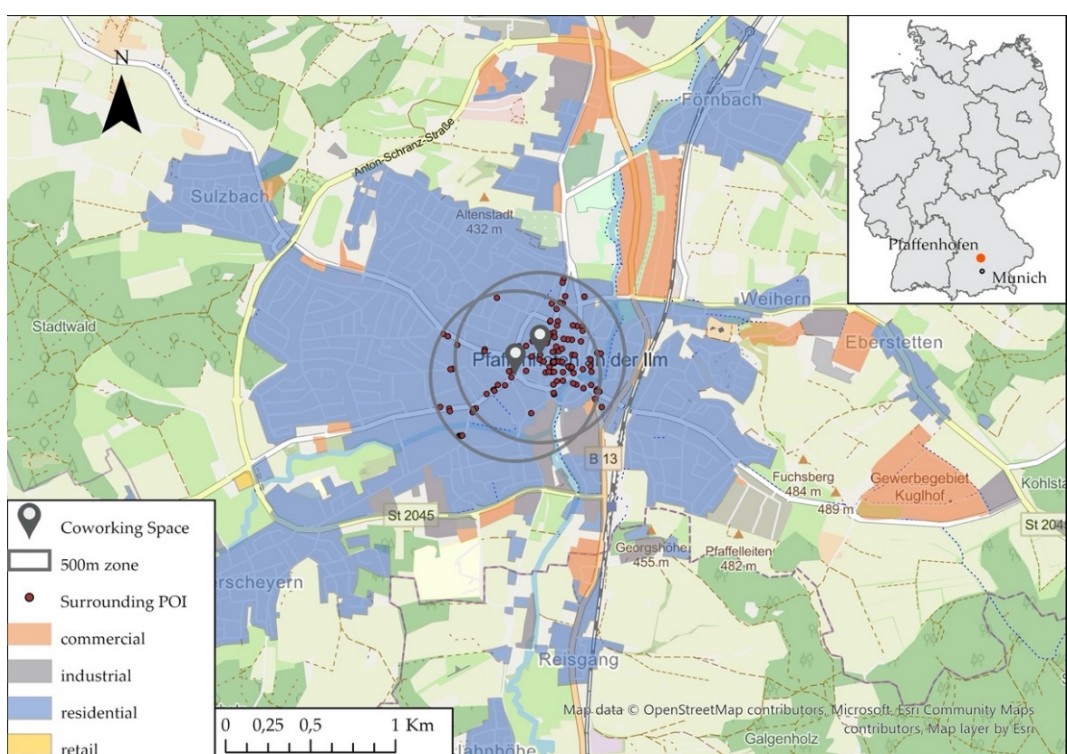

**Figure 11.** Map of POIs as listed in Table 2 in a radius of 500 m around coworking spaces (CWSs) in Pfaffenhofen an der Ilm (without scale) source: ArcGIS® software by Esri, OSM, gefabrik.de, coworkingmap.de, © GeoBasis-DE/BKG (2020).

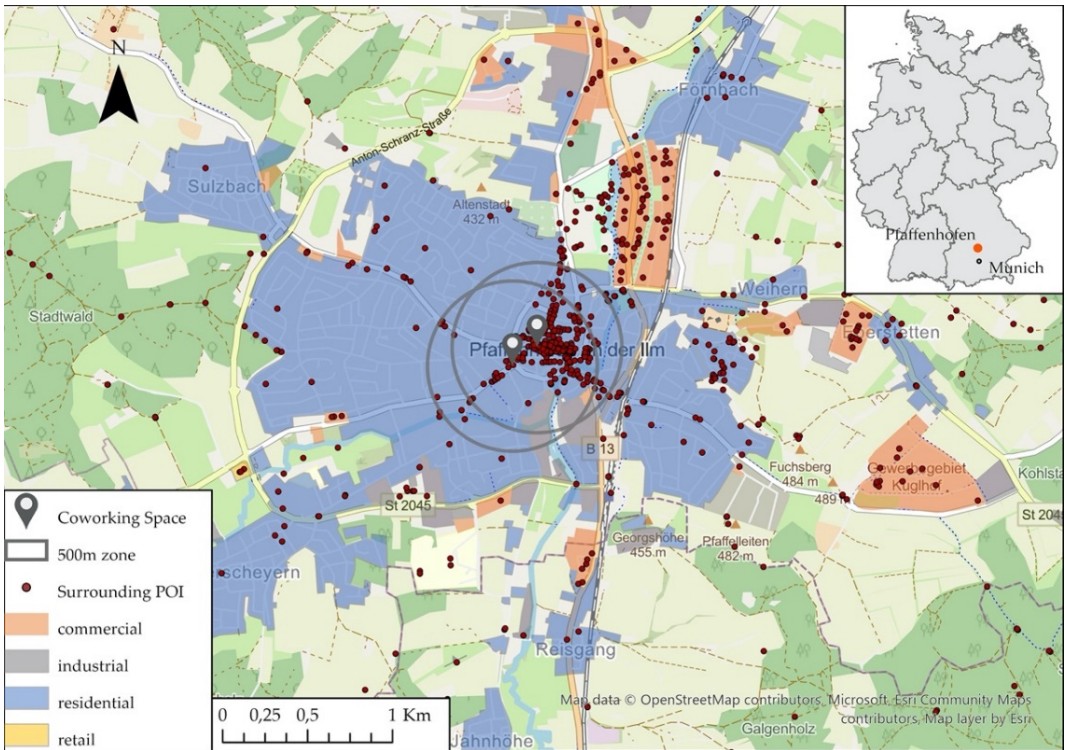

**Figure 12.** Map of all POIs of the category 'pois_free' in Pfaffenhofen an der Ilm (without scale) source: ArcGIS® software by Esri, OSM, gefabrik.de, coworkingmap.de, © GeoBasis-DE/BKG (2020).

*4.5. Comparison of POIs in the Vicinity of CWSs to POIs in the Vicinity of Other Places of Work*

As described above, we found a high number of POIs, which are relevant in the course of a regular working day, in the vicinity of coworking spaces, especially if they are

located in non-peripheral regions and in areas that are dominated by residential or commercial land use. To compare this with other, more traditional locations of white-collar office work, we took a look at office locations in both of the regarded cities, Munich and Pfaffenhofen an der Ilm. For Munich, we chose the office city/city of offices Unterföhring, Dieselstraße (Figure 13), which is dominantly used as a location for offices, with companies as Allianz, ZDF, Pro7Sat1 (television broadcasting companies). For Pfaffenhofen an der Ilm, we chose the location of the company Hipp GmbH in Georg-Hipp-Straße (Figure 14), which is a huge and important employer in Pfaffenhofen an der Ilm. As coworking space, for example, we chose the coworking space EchtLand in Pfaffenhofen an der Ilm and MATES in Schwabing, Munich.

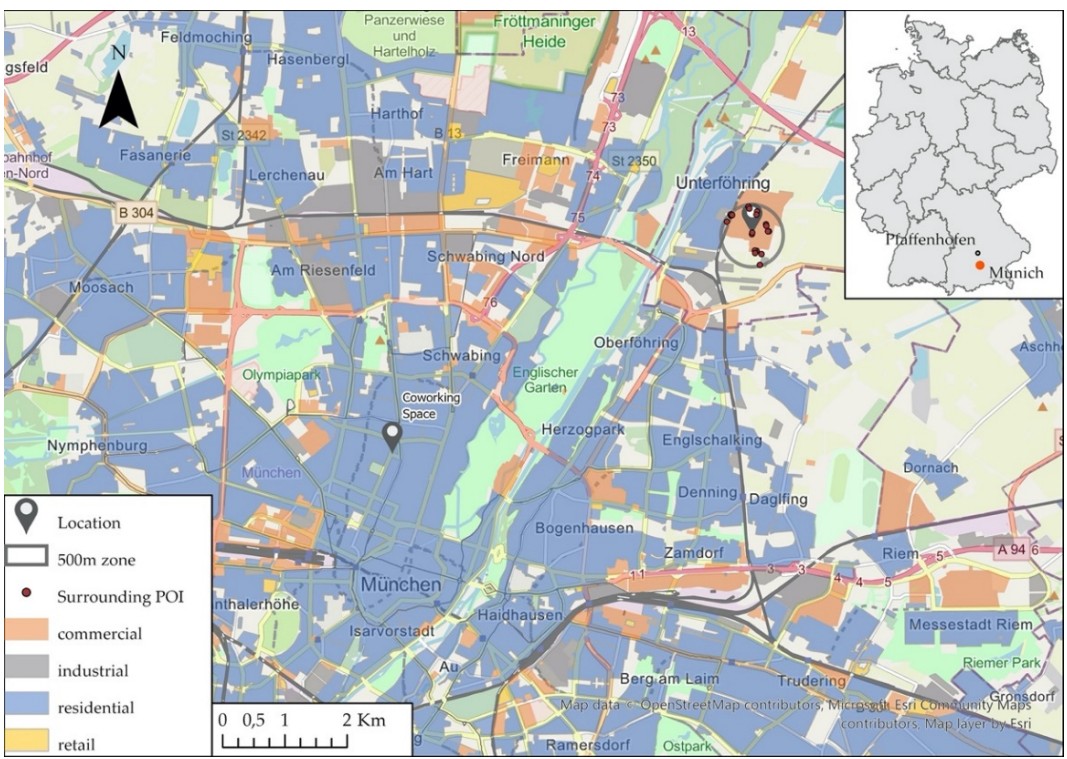

**Figure 13.** Map of all POIs in the category 'pois_free' in Unterföhring, Munich (without scale). Source: ArcGIS® software by Esri, OSM, gefabrik.de, coworkingmap.de, © GeoBasis-DE/BKG (2020).

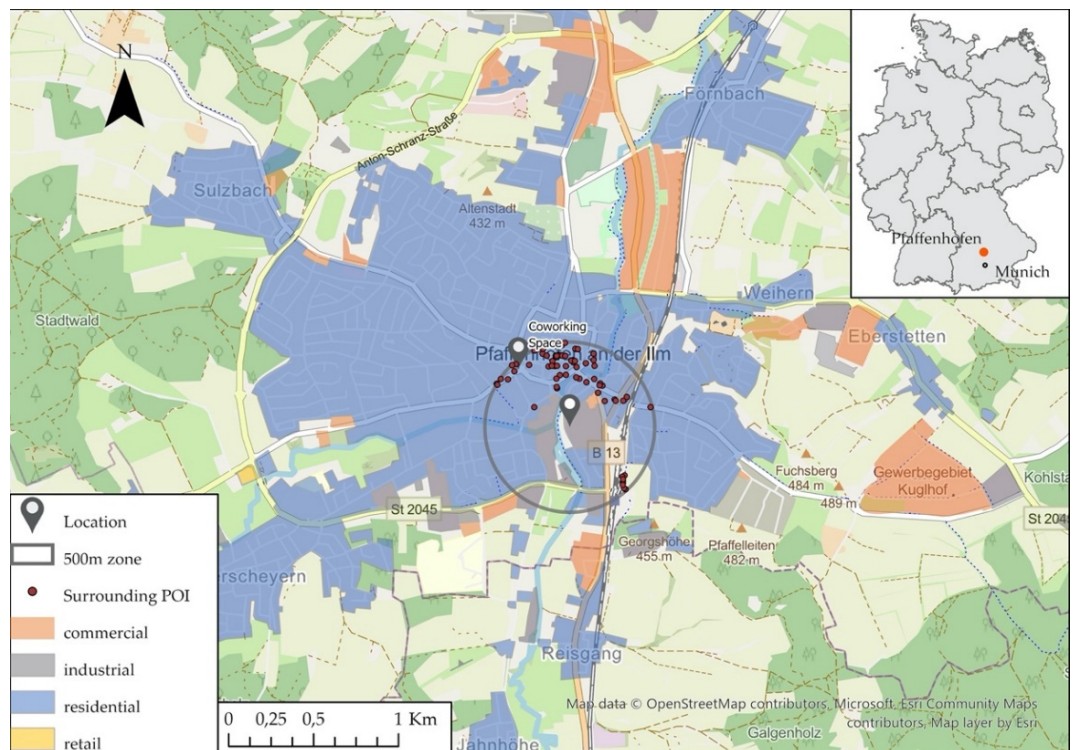

**Figure 14.** Map of all POIs of the category 'pois_free' in Pfaffenhofen an der Ilm, Georg-Hipp-Straße (without scale) source: ArcGIS® software by Esri, OSM, gefabrik.de, coworkingmap.de, © GeoBasis-DE/BKG (2020).

From Figures 13 and 14, we received a first impression of the number of POIs in the vicinity of the office locations. To obtain a clearer picture, we analysed the number of relevant POIs in the vicinity, again within a radius of 500 m.

By analysing the POIs in the vicinity of these different locations (Figures 15–18)—office cities, on the one hand (Figures 15 and 17) and, coworking spaces on the other hand (Figures 16 and 18)—we found in general a higher number of POIs close to the CWS, especially in non-peripheral regions (Figure 17).

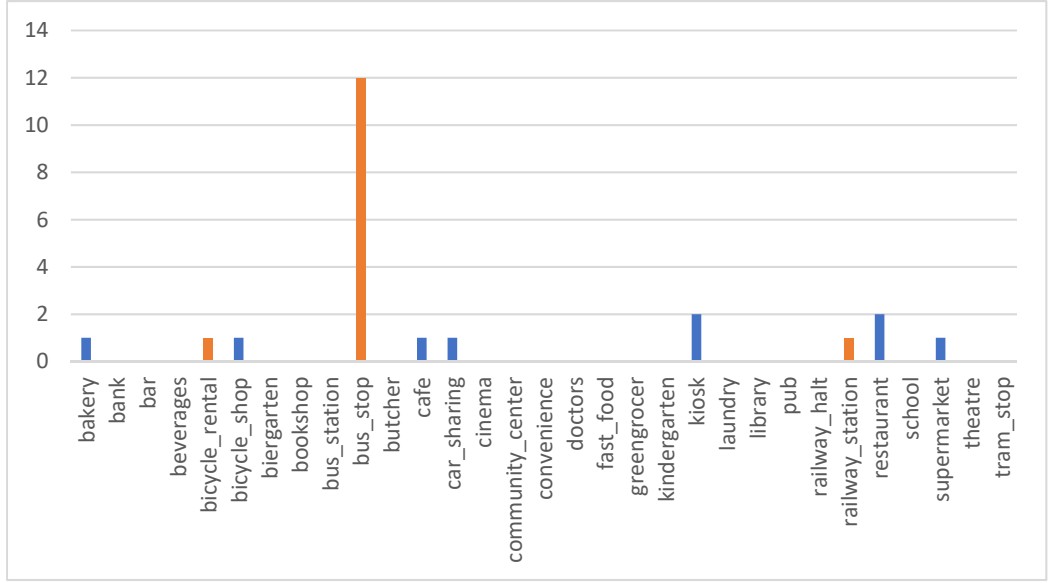

**Figure 15.** Number of POIs in the vicinity of 500 m in the surrounding of the office location Dieselstraße in Unterföhring, Munich.

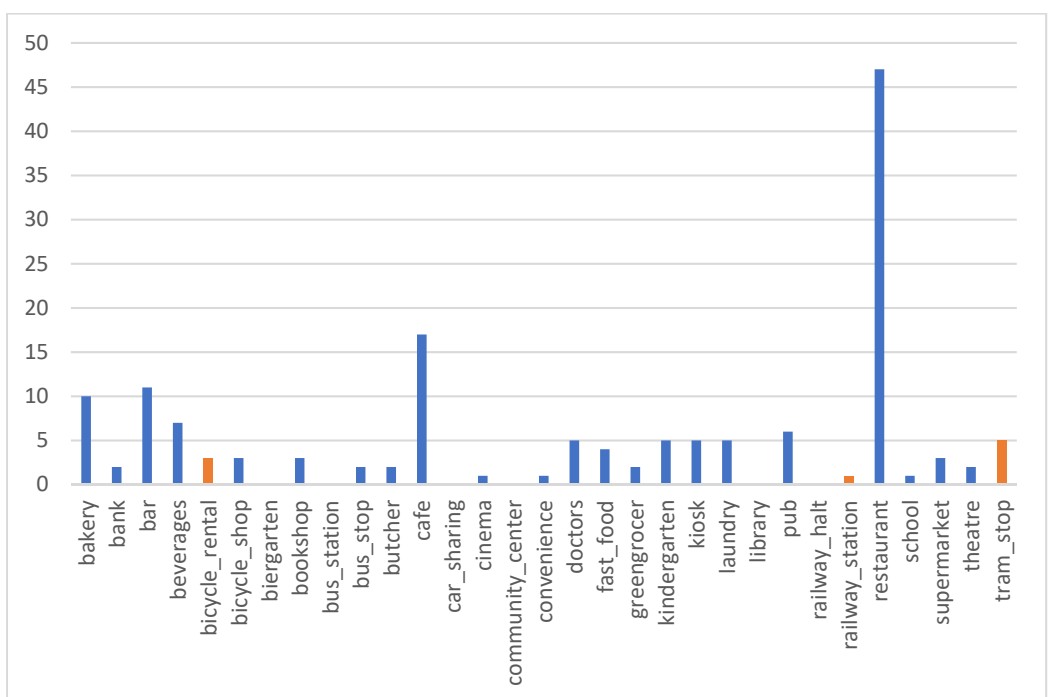

**Figure 16.** Number of POIs in the vicinity of 500 m in the surrounding of the CWS MATES in Schwabing, Munich.

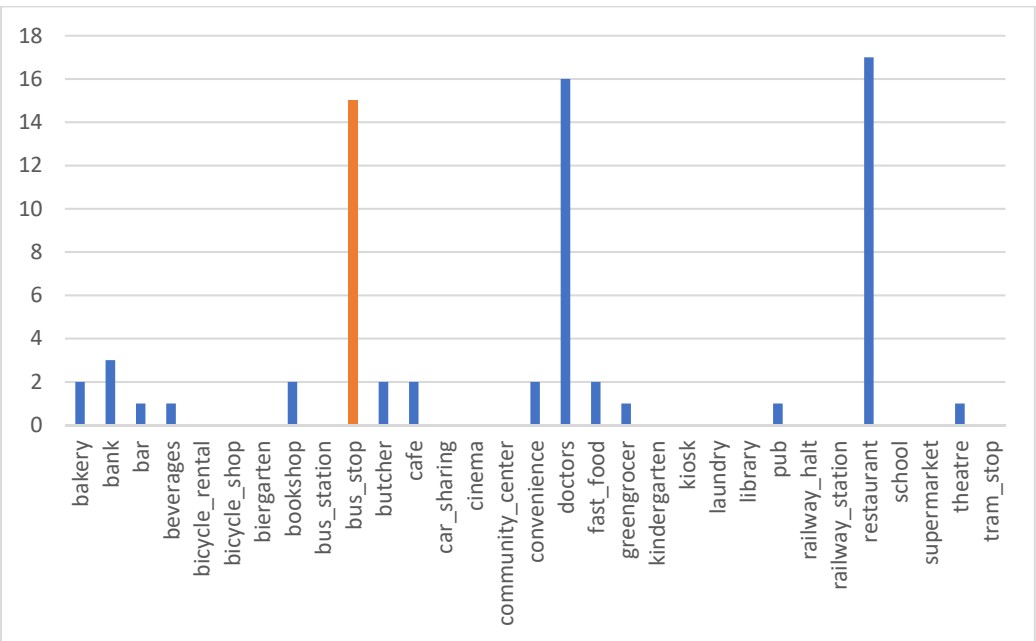

**Figure 17.** Number of POIs in the vicinity of 500 m in the surrounding of the company HIPP in Pfaffenhofen an der Ilm.

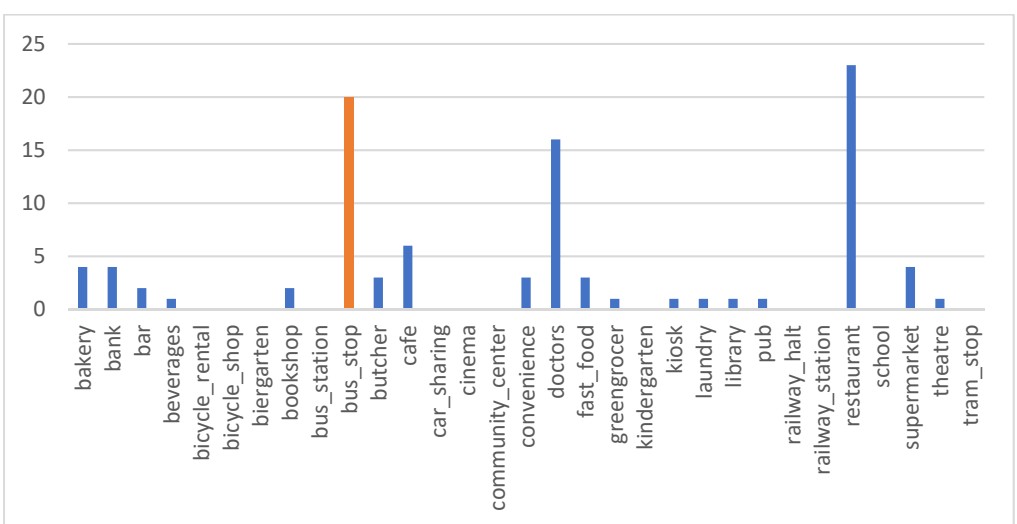

**Figure 18.** Number of POIs in the vicinity of 500 m in the surrounding of the CWS EchtLand in Pfaffenhofen an der Ilm.

The high number of "bus_stop" POIs in the vicinity of the companies, compared with the vicinity of CWSs, is noteworthy; furthermore, the number of POIs in general in the vicinity of Dieselstraße is more or less at the same level to the number of POIs in the vicinity of the company HIPP. It seems that CWSs are more likely to be located in a more urban, mixed-use surrounding, and also when they are located in a peripheral/rural region.

## 5. Discussion

### 5.1. Peripheral and Non-Peripheral Location of CWSs

The results demonstrate that the majority of coworking spaces (CWSs) represents tertiary work [106], non-physical work, knowledge-based work, creative work [63] or freelance work [67]. This type of work mainly occurs in non-peripheral regions (562 CWSs in non-peripheral areas—79%, 149 CWSs in peripheral areas—21%). Hereby, we employ the definition of non-peripheral areas according to Workgroup 1 of the COST action CA18214, as functional urban areas (FAU) below 200,000 inhabitants [131]. With this, we have an estimate of where coworking spaces are mainly located, which is still in non-peripheral, more urban environments.

This contradicts the findings in Italy, where the CWSs are in 76% of the cases in urban areas, in 5% of the cases in intermunicipal areas, in 16% of the cases in outlying areas, and in 3% of the cases in intermediate areas. CWSs apparently hardly exist in peripheral areas and ultra-peripheral areas [101].

In the last years—and presumably fuelled by the COVID-19 pandemic and the increase in remote work—the benefits and spread of CWSs in rural regions have been widely discussed [80,96,97] and politically supported [84–86]. The type of location where CWSs are located seems to depend on their concept, e.g., Retreat, Coworkation, Commuter Port, etc. [80]. With political support [86] and possible funding [82], it can be assumed that the spread of CWSs in peripheral regions will increase.

### 5.2. Location of CWSs and Land Use

The above-described analysis gives only a very rough cognition, where coworking spaces are located. With the land-use classification from the OSM dataset, we can identify the dominating use of the neighbourhood in which a coworking space is located. Here, we found a dominance of CWS locations in 'residential' neighbourhood by 63% in general related to other kind of land use. There is a dominance of 'residential' land use of 62% in non-peripheral and a higher share of 69% 'residential' land use in peripheral regions.

Residential neighbourhoods provide a high nearby potential of users or customers for CWSs and for other amenities, offers and services (POIs) [49]. A spatial close relationship between POIs with each other and with CWSs makes it more likely that multipurpose trips will be taken [127,128].

While in non-peripheral regions, the share of CWSs located in 'commercial' neighbourhoods is at 22% and for 'industrial' areas at 6%. In non-peripheral neighbourhoods, the share of CWSs in 'commercial' neighbourhoods is at 11% and for 'industrial' at 12%, which doubles the result compared to non-peripheral locations. The higher share of CWSs located in 'industrial' neighbourhoods remote from—or not inside of—'residential' neighbourhoods makes it more likely that people—users or tenants of CWSs—will travel to work by car, due to the higher distance [22] and the time saving and faster mode of transport by car. The separation of town districts by function as it was proposed by the idea of "Garden City" [12] and the "Charter of Athens" [27]—which were reasonable in previous times—leads to a higher average distance between the place of work, residence and other destinations—in our research, the regarded CWSs and POIs.

If commuting to the regular—not necessarily daily—place where work is performed is by car, it is more probable that other daily trips, such as going to buy groceries, going to sports or recreation facilities, is carried out by car as well [127]. This will inevitably lead to a higher number of car trips, which is more evident in peripheral regions, where the distances for daily trips are higher [22]. If a CWS is located in an industrially or commercially dominated area, it is also more likely that people continue to rely on car transport for daily trips. For the environment, this would lead to higher emissions of $CO_2$/greenhouse gases, and for infrastructure planning, this would ultimately lead to higher demands for parking and road space as well as higher costs for road maintenance. Such a development is contra-effective for the goals of sustainable transport (as recommended by the United Nations Secretary-General's High-Level Advisory Group [46]), Sustainability Strategy of Germany [47], the National Platform Future of Mobility [48], and the concept of the 15-Minute City [118].

The 'New Leipzig Charter' planning policies adopted on a national level [47,86] and EU level formulate the aim D.1.1 Active and strategic land policy and land use planning "Polycentric settlement structures with appropriate compactness and density in urban and rural areas with optimal connections within cities to minimise distances between housing, work, leisure, education, local shops and services" [121].

Strategies that take this focus on vicinity into account have not been implemented much so far; more traditional functionally separate structures, which are legally specified by the framework (BauNVO), are adopted by municipalities as land-use plans. Hereby, the land consumption often exceeds the population growth [115].

If a CWS is located in the centre of a town or village, it could give an abandoned house or shop a new assignment; maintain the already built grey energy; bring vividness and spending capacity to the traditional town centre, with amenities, shop, services (POIs); and prevent people from driving to the outskirts by car. Admittedly, this is not guaranteed, but it is more likely if it is more attractive. Following the New Leipzig Charter and regarding our findings on the relevance of vicinity, the functional separation of land use through the BauNVO should be questioned.

From a legal perspective, a CWS can also be approved in residential, retail, industrial or commercial areas, in which CWSs can also be found. In residential areas, there is a higher number of different uses and diversity recognizable in the higher number of POIs. This grade of diversity seems to be more attractive for CWSs as there can be more CWSs. The diversity of city districts is what Jacobs was aiming at 60 years ago. The Charter of Athens and the BauNVO ultimately prevents city districts from being diverse, i.e., having different uses, not allowing only certain uses and excluding non-listed uses, for the BauNVO [32]. The regulations of the BauNVO closely connected to how the Charter of Athens approaches the separation of uses. The Charter of Athens relies on the idea of the Garden City with functionally separated districts [32]. Additionally, 60 years ago, Jane

Jacobs promoted pedestrian-friendly cities [53], which is currently taken up in the New Leipzig Charter or the 15-Minute City. Such a pedestrian-friendly city district could generate residential areas with a high number of POIs and thereby a diverse range of amenities, offers and services. Those amenities in residential areas attract more CWSs than in other areas.

*5.3. POIs and Their Spatial Relation to Coworking Spaces*

The sample of daily trips consists of work-related and more private occasions, with nearly one-third being related to education and work, one-third to trips for leisure issues, and one-third to shopping and private errands [22]. If work is performed in a coworking space is relevant, if there are destinations for other purposes, as mentioned above, close to the CWS, to combine the trip to or from the place of work with the trip to or from the coworking space [127].

The POIs we have chosen and listed in Table 2 can be regarded as potentially related or combinable with the trip from or to the place of work. Due to the relationship between the place of work and the listed destinations, the spatial proximities of the place of work and other daily destinations are indicators for a relevant accessibility of these, especially when they are in easy walkable distance of maximum 500 m (5.7–6.4 min) [124].

From the regarded POIs classes (Table 2), in total ca. 700,000, nearly one-twentieth is located in the 500 m radius around CWSs. Around 88% of this share is located in the 500 m radius around CWSs in non-peripheral regions. The density of POIs in non-peripheral regions is much higher than in peripheral regions, which can be assumed.

Taking a general view on peripheral and non-peripheral regions, we found a high share of POIs 'bus_stop' (8.1), 'café' (7), 'fast_food' (6.2) and 'restaurant' (14) in the 500 m vicinity of an average coworking space. The numbers are higher in non-peripheral and lower in peripheral regions, reasoned in the general difference of POI density. The higher density of relevant POIs in non-peripheral regions was expected, but it underlines the relevance of a high number of POIs spatially related to the place of work [127,128].

When we separated our consideration of the type of land use, we found a clear spreading of POIs between the categories 'retail', 'residential', 'commercial' and 'industrial'. It should be noted here that 'retail' can be both inner-city locations and shopping centres on the outskirts of settlements.

The POI 'restaurant' seems to be relevant as an option to buy lunch during the course of the day and take a break from work. Regarding the number of 'restaurant' POIs in non-peripheral regions, in the land-use categories, we found an average of 28.9 in 'retail', 22.2 in 'residential', 9.7 in 'commercial' and 2.1 'industrial'. The availability and range of offers is high in 'retail' and 'residential' areas and attractive as a location for a coworking space, which seldom have their own lunch service.

Regarding other highly relevant destinations for daily or regular trips [119], we take a closer look at the POIs 'supermarket' and 'kindergarten'. In non-peripheral regions, we found 3.1 'supermarket' POIs in 'retail' areas, 3.3 in 'residential' areas, 1.6 in 'commercial' areas and 1.0 in 'industrial' areas 500 m around a coworking space.

In peripheral regions, the situation deviates outside of 'retail' areas. Here, we found 3.2 'supermarket' POIs in 'retail' areas, only 1.2 in 'residential' areas, only 0.4 in 'commercial' areas and 0.2 in 'industrial' areas 500 m around a coworking space. We found a slightly higher number of 'supermarket' POIs in 'retail' areas in peripheral regions, and in non-peripheral regions, a massive drop in 'supermarket' POIs in 'residential', 'commercial' and 'industrial' areas 500 m around a coworking space.

In non-peripheral regions, we found 1.5 'kindergarten POIs in 'retail' areas, 2.3 in 'residential' areas, 0.8 in 'commercial' areas and 0.6 in 'industrial' areas 500 m around a coworking space. In peripheral regions, we found 2.5 'kindergarten' POIs in 'retail' areas, 0.4 in 'residential' areas, 0.2 in 'commercial' areas and 0 in 'industrial' areas 500 m around a coworking space. The availability of a kindergarten should be more important in a residential area than in an industrial area, which reflects our findings. It is interesting to

note that in peripheral regions, the value for retail areas is higher than for residential areas. This could be due to the small-scale character of rural towns, where central areas are more likely to be attributed to shopping areas. However, this is not the case for our sample communities and should be considered further in future research.

It seems to be significant that CWSs in peripheral areas have more POIs in their vicinity than in non-peripheral areas. This suggests that one could potentially increase the attractiveness of CWSs in non-peripheral regions by increasing the number of POIs (Figure 16).

It is, however, important to note here that a high number of POIs need to be relevant for daily use, as indicated by the list in Table 2. In this way, it is possible to combine necessary trips with the trip to the job, and to reduce the commuting time, avoid traffic jams, decrease $CO_2$ emissions, and support the local economy. CWSs and other amenities represented by POIs such as shops and services can benefit from each other. On the one hand, the presence of CWSs enables the increase in potential users of CWSs to utilise the services of POIs. On the other hand, users of a CWS bring purchasing power and customer frequency to the offers in the vicinity of the CWS. In addition to the benefits for users, POIs also offer the opportunity for cooperation and networking, not only within the CWS but also with the neighbouring POIs [81,90]. CWSs and POIs, e.g., restaurants, shops, cafés, and cultural institutions, could cooperate, enrich the respective offerings and provide opportunities for network expansion.

With a higher visitor frequency, decaying inner towns could gain vitality and avoid or reduce the donut effect [60,61].

## 6. Limitations

Although the findings of this research have generated a first insight into the spatial and thematic relations between the place of work and the place of residence as well as the essence of why and how people use coworking spaces, we also realise that the research approach was not without limitations. First of all, we had to rely on open-source data, which may not have been validated at all times. Secondly, we made a number of assumptions in our modelling, such as walkability distance. Obviously, such distances could be further detailed with topographic height and steepness information, for example, complemented by pedestrian surveys to test the degree of walkability or carry out an accessibility analysis with routing algorithms using a topological, routable road network in GIS. However, this was not the main purpose of this specific study. The first step was to find general trends on spatial relations and finding relevant indicators. Thirdly, one could also debate the choice of POIs. The large variation in identified POIs of specific land-uses between peripheral and non-peripheral regions suggests that the land uses recorded in the OSM database are less comprehensive and precise, at least in more rural areas. We found that the geocoded locations of CWSs in ArcGIS based on the address can produce deviations in a few cases.

With more cases of coworking spaces in and outside of Germany, the picture we produced with our research could be improved and maybe generalised. This could provide more insight into the consistency of results. Constructing detailed spatial models to carry out simulations could predict future developments. Results could be validated by remote sensing in order to find whether one can detect, and possibly automate, the dynamic relations between work and residence. Surveys on the behaviour of users, tenants and operators of CWSs could give a clearer picture of changes in the course of the day and usage in time, money and presence.

We excluded 12 CWSs from our research because they have no POIs in their vicinity. Reasons for the lack of POIs could be the remoteness of these CWSs. This could be the case for CWSs that are used more for retreats or 'workations' [80]. The background could be further explored in future studies.

Even if the frequency of specific POIs is particularly high in the vicinity of CWSs, by our judgement, no explicit requirement for a specific POI can be identified. The higher

frequency of POIs only seems to make a location attractive for CWSs in principle, as they occur more frequently here. However, this study cannot make any statement about the economic success and thus the long-term existence of the CWS.

A closer look at prototypes of CWSs in further research could investigate in specific circumstances, activities, business models, etc.

### 7. Conclusions

The spatial analysis of POIs in relation to the location of coworking spaces confirms that coworking spaces are more likely to be located in non-peripheral areas than in peripheral areas. In fact, our findings reveal that 79% of the examined cases were located in non-peripheral areas and 21% in peripheral areas. However, the review of the variety of services and the connection of this variety of services to CWSs reveals novel insights in the discourses so far.

First of all, the vicinity of CWSs can be described in 62% of the cases as residential, 22% as commercial, 6% as industrial, and 8% as retail for non-peripheral CWSs. In contrast, for peripheral areas, the vicinity reflects a surrounding which is in 69% of the cases residential, 11% commercial, 12% industrial, and 4% retail. Hence, in non-peripheral areas, there is a larger variety and more balanced distribution of services in the vicinity of the CWS, suggesting that the more peripheral an area is, the more variety and more equal distribution there may be. Secondly, there is a clear relationship between the types of services and the attractiveness of CWSs.

A CWS is more attractive if it has easy access to a high number of relevant POIs. This implies that when launching a CWS, one has to take both the variety and type of additional services into account.

Thirdly, an important consequence of establishing vibrant CWSs is that it may create and foster local vitality and versatility in the region and contribute to a more attractive quality of life. There is still a separation between private life and working life, yet this separation is relatively small in terms of time and distance. The direct effect is that the number of trips can be reduced drastically, but an indirect effect is that being more engaged in a certain surrounding will also have an impact on the sense of belonging and identity. This fuels the allocation of spending capacity in the vicinity of CWSs, enhancing lifelines and vitality of the public space surrounding the CWSs, which should be located in the inner-town. This is even more important in rural regions, where distances and daily trips are usually longer than into non-peripheral regions and urban areas.

**Author Contributions:** This manuscript is a part of M.H.'s ongoing research. Conceptualization, M.H.; methodology, M.H. and K.-H.K.; software, K.-H.K.; writing—original draft preparation, M.H.; writing—review and editing, M.H. and W.T.d.V.; visualization, K.-H.K.; supervision, W.T.d.V. All authors have read and agreed to the published version of the manuscript.

**Funding:** This research was funded by TUM open access publishing fund.

**Institutional Review Board Statement:** Not applicable.

**Informed Consent Statement:** Not applicable.

**Data Availability Statement:** Data available in a publicly accessible repository that does not issue DOIs. Publicly available datasets were analysed in this study. These data can be found here: Coworking Spaces: www.coworkingmap.de (retrieved on 08 April 2021), OSM-Data: http://download.geofabrik.de/europe/germany.html (retrieved on 4 June 2021), © GeoBasis-DE/BKG (2020) https://gdz.bkg.bund.de/index.php/default/open-data/verwaltungsgebiete-1-250-000-ebenen-stand-31-12-vg250-ebenen-31-12.html (retrieved on 14 October 2021).

**Conflicts of Interest:** The authors declare no conflict of interest.

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
