# Peer review of "Location of Coworking Spaces (CWSs) Regarding Vicinity, Land Use and Points of Interest (POIs)"

_land, doi:10.3390/land11030354_

Round 1

Reviewer 1 Report

Abstract

  • 1-Despite the reformulation of the abstract, please clarify the following sentence (line 14-15) “The place of work is, besides the place of residence, a main travel destination in the course of the day”.
  • 2-To align with the content proposed in the special issue of Land, the authors are suggested to frame the research within current programs and rural development strategies.
  • The use of 13 keywords is not justified. It is requested to adjust the number and format of the keywords to the criteria of Land.

Introduction

  • This introductory part is well focused, and a great work of data collection and spatial analysis is demonstrated, although the section falls short in explaining the relevance of this work and its contribution to the literature.
  • In summary, current research focuses on the analysis of CWS and the supposed disconnect between the places of work and residence or life of the population: we have a different opinion on that question. While the subject is challenging, certainly the literature the authors use to justify the objectives and approach used seems biased. Everything seems to be aimed at arguing a reality that exists in the Western world, and more specifically in Central Europe, but there are many articles that deal with this issue in other areas of the planet, with results that can be used to refute the ideas presented. If you want to address this interesting topic from a perspective of change like the current one, it is suggested to be more specific about how the research proposed is filling a void (it is not enough to ask some research questions and leave them there).

Literature review of legal, social and historical context of separation between work and residence

  • Part 2, on Literature review of legal, social and historical context of separation between work and residence, does not seem very justified in the text, at least with the formula chosen by the authors. First, I consider that a good part of the determinations in this section could be ascribed to the introductory part, and the division into two parts is not necessary. On the other hand, the approach in this section is a bit hectic, preventing the reader from understanding who the actors are and the precise focus of the article, its objectives and questions. Above all, the most relevant thing is that the reader concludes the reading without having an idea about the authors' approach to the relationship between places of work and places of residence.
  • Neither section 1 nor section 2 justifies clearly and sufficiently why Germany was chosen to carry out the study.
  • In line 243, the reference to Ridwan and Dimas is not well referenced.
  • The legal, social and historical approach that is proposed in the section seems to forget the importance of the territorial approach in an analysis of the relationship between the human being and how it is linked and related to the environment that inhabits and works. It is suggested to approach the issue from a perspective that is closer to the concept of economic and regional geography, developed through approaches and models of rural development and their evolution in the world, e.g., European policies and rural development programs and rural development strategies; Integrated forms of urban – rural planning and multilevel governance, e.g., urban partnerships and the role of the city in the development and stability of the rural population

Research design

  • Section 3, research design, may be part of the same section on methods and materials, thus following the criteria of MDPI and Land.

Methods and materials

  • In line 290, the "Theoretical concepts, ideas and models insufficiently capture the current realities of remote work and coworking" is not sufficiently justified. There is specific bibliography on the subject, abundant and current that it is recommended to review.
  • On line 327-328 the database www.geofabrik.de is mentioned as the data source for the localization. More information is requested on the use of this database, as well as a critical analysis of it and a justification for its incorporation into the research.
  • Figure 10 is very ambiguous and, in my opinion, quite unnecessary. The result sought with figure 10 can be achieved with small position map figures in figures 11, 12 and 13. In addition to this, it is noted that the geographical and cartographic quality of figures 10 to 16 it is very poor. The non-existence of scales, north bars, and other key elements in cartography is noted.

Discussion and conclusions

  • The overall text seems still too confusing. You did not introduce a clear problem, why you were doing it and what it brought to the discussion on land uses. The fragmentation of the county decreased, fine, but how and why did it decrease? What are the consequences on the landscape and the importance of this harmonization of the territory? You want to demonstrate the importance of socio-economic factors over land use and its evolution, yet you are not defining them and linking your quantitative data to a qualitative analysis that would highlight your argument. The elements to demonstrate such an argument are probably in your hands, but your ways of explaining and formulating your sentences / narrative negatively impact the quality of the paper.

Author Response

Thank you for your comments. Please see the attachment for the reply (reviewer 1). 

Reviewer 2 Report

The submitted article has been improved in terms of form and content. However, a complete revision of spelling, punctuation and grammar is suggested to the authors, as there are some errors.

For example:
line 16, there is a full stop in front of "are".
line 16, please, clarify the aim of the article, it is not well redacted
Line 85-86, specifically is repeated
Line 89-91, "the following sections" is repeated.
Line 499, please, avoid repetition of words
Line 573, please, correct peripherical
Line 691, there is a loose "d".
Regarding the cartography, it is recommended that figure 10 be inserted, in a small box, in each of the following figures (11,12,13, 14), and not as an individual map, as it takes up too much space and has too little information.

The discussion remains very brief, consisting only of a further explanation of results. If there are no previous studies on CWS, a good idea would be to relate the results to accessibility studies, urban flow studies, or to the implementation of spatial planning policies, both at municipal and federal level. On this point, the discussion section is very poor and does not yet resolve the main weakness of the article.   

Author Response

Thank you for your comments. Please see the attachment for the reply (reviewer 2).

Reviewer 3 Report

Overall, it is a timely and interesting topic to explore. Please have it proofread as there are quite a few typos and sentences where English grammar is not correct. I am missing the „so what" bit in the discussion and conclusion – what do all the results mean? What conclusions can be drawn for practice? For instance, can we say that after exploring the different POIs by type of land use in periphery and non-periphery, there are certain POIs as the main decision-making factors for operators to establish coworking spaces these?

Couple of specific comments:

  • When you say „Which and where are services and offers located in their surrounding of CWS, that can be relevant for users and tenants?” do you mean as relevant in attracting tenants/ users to those particular cw spaces? Or relevant as in spaces that could add value to the area therefore making it more appealing for people to joint those particular coworking spaces just because they are located in certain areas where lots of different opportunities/ spaces can be found?
  • Definition of peripheral/ non-peripheral. According to this sentence, peripheral is not only the area where the population is smaller than 200,000 but also where the population is bigger than 200,000 inhabitants. Please make it clear because these 2 cannot apply in the same time to a location. „The location of a coworking space is thus classified as “peripheral” attribute as follows: (a) 1, if the coworking space is located outside a metropolitan region. Or if they are located within the metropolitan region and this region is smaller than 200,000 inhabitants, (b) 0, if the spaces are located within a metropolitan region with a population greater than 200 000 inhabitants.”
  • „12 coworking spaces do not have a POI in their vicinity and were therefore excluded from the calculation.” I understand why you left out from the sample but wouldn’t it be worth to explore the area to understand what might have been the reason for the existence of those coworking spaces? Not as part of the article but in general.
  • The part on p7 where you talk about POIs is hard to understand. I suggest you share only a digestible amount of information. Did you finally select 41,166 POIs or 56,422? You then didn’t explain which were the selected POIs you kept for the analysis, only put a table that lists some of them. Can you please make it clear in the Methods and Material section? You also don’t talk about the type of land use here, only mentioning it in the Results section. I think it would be worth to introduce them in this section.
  • - Results: I don’t think a merged figure (Figure 2 ) is necessary if you later talk about peripheral and non-peripheral areas separately. You could also make a point on the retail %: usually in retail, you have all the amenities needed for a coworking spaces in one place (café, restaurant, parking, toilet, various shops, etc) which are also appealing for people so they could spend all their time, but also for operators for whom it’s easier to just go to the retail shop and bring in a coworking space since all the amenities are in one place. On the other side, many retail is struggling and so, coworking spaces could support the regeneration of the broader area by adding extra services (workplace) for the retail.
  • - Figure 6 and 7: I’d split between peripheral and non-peripheral results and illustrate them in separate figures. When you explain what could these figures mean, wouldn’t make more sense to round the numbers? 8.8 fast food restaurant around the coworking? It doesn’t make sense. I’d rather say 8-9 fast food as an example.
  • Is Table 3 the same as in the above figures just in number format? If you don’t explain what we can see on figures / tables in the text, I suggest to remove.
  • Figure 10-14: nothing is explained what are those, why are they there, how are they connected to the article. Please explain in text. I kind of understand you want to illustrate the above analysis on reference cities but please explain how does it contribute to your findings and conclusions
  • - In the Discussion section please explain what each of your finding means rather than repeating what was said above. „In non-peripheral regions we found 1,5 POIs ‘kindergarten in ‘retail’ areas, 2,3 in ‘residential’ areas, 0,8 in ‘commercial’ areas and 0,6 in ‘industrial’ areas 500 m around a coworking space. In peripheral regions we found 2,5 POIs ‘kindergarten in ‘retail’ areas, 0,4 in ‘residential’ areas, 0,2 in ‘commercial’ areas and 0 in ‘industrial’ areas 500 m around a coworking space” – what could this mean?
  • - „coworking spaces are more likely to be located in non-peripheral areas” – this is a valid conclusion but why? How does your findings on POIs contribute to this? Could we say that because in cities we have more amenities than in rural places? What are the most common POIs in these different areas? Can we come up with a list of must-have POIs for coworking spaces to succeed in periphery and non-periphery areas?

Author Response

Thank you for your comments. Please see the attachment for the reply (reviewer 3).

Reviewer 4 Report

This paper sets forward a target to relate the location of Coworking Spaces (CWS) to landuse structure and Points of Interest, adopted by OSM, using GIS analysis.

However, the connection of methods used to urban planning theory is trivial and incomplete. Introduction refers to very old archetypes of urban form like Garden City, Broadacre City etc while in a contemporary research the relation between work and residence could not be based simply on models like the Charta of Athens. The authors in many parts of the text do not justify their comments, (e.g. that the number of inhabitants is shrinking). Moreover, even the key idea of this article seems incomplete: CWS are just projected on a map and related to other data, yet, POIs are used without any hierarchical  classification based on their importance. Results seem rather obvious: a relative higher amount of specific POIs is observed in the vicinity of CWS, however, there is no novelty in the methodological approach, nor, in the theoretical concepts applied here. The authors "blend" various different concepts of urban planning theory like 15-min city, Charta of Athens, even Jane Jacobs theories, in way that is not appropriately structured, nor justified.

i suggest that this work should be fully re-organized, base on more concrete spatial analysis methodology, and trying to refer to only recent literature on commuting patterns and the role of CWS in structuring the economic as well as spatial elements of the city. A more focused case study analysis in a single selected city would help to identify more concrete results, because using the whole German State as reference could be misleading and providing results that are very "average".

Author Response

Thank you for the comments. Please see the attachment for the reply (reviewer 4).

Reviewer 5 Report

The manuscript is interesting and innovative in defining the location of coworking spaces and their uses.
Some minor changes that could be added would be:
1.- Explain why 12 coworking spaces were not considered in a higher way. The comment is that these do not have points of interest, but they could be considered as well. 
2.- Justify why 500 metres, and not other distance in a better way.
3.- Reduce the number of figures, and describe these deeper in the text.
4.- Could describe the differences of CWs in peripheral and non-peripheral areas. What kind of activities and functions in both types of areas?
5.- In a table, insert the predominant activities and differences between CWs in land use, location and POIs, for peripheral and non-peripheral areas. 
6.- Revise some orthographic mistakes: page 2, line 77: "and the and the"; page 27, line 699: "und".
7.- Contributions of CWs for rural and regional development. These offices could help or support the territorial development?: reduce the "donut" effect in city centres or attract new settlers in rural areas. European Network of Rural Development shows good practices of CWs and their importance for rural development.
8- Finally, could you show "type" cases of CWs for peripheral and non-peripheral areas?.

Author Response

Thank you for your comments. Please see the attachment for the reply (reviewer 5).

Round 2

Reviewer 1 Report

The proposal has improved remarkably after the review rounds. We thank the authors for having taken into consideration the suggestions that, in the case of this reviewer, have been made in the sense of formulating nuances, suggestions and corrections raised with the exclusive and praiseworthy desire that his research has a greater analytical scope and, therefore, a positive and quality impact on the scientific community. sure they will know appreciate useful considerations of the evaluators.

Reviewer 2 Report

The comments have been solved

This manuscript is a resubmission of an earlier submission. The following is a list of the peer review reports and author responses from that submission.

Round 1

Reviewer 1 Report

This study reported on the location of coworking spaces (CWSs) and their geographical association with other POIs. Although it dealt with an important topic in envisioning the city after the COVID-19 pandemic, it is inadequate for publication in this journal, as described below.

Major comments

L.118. The objectives are exaggerated in relation to the research content. The objectives should be stated in line with the research questions.

L.122-126. Three research questions should be reconsidered:

  1. It seems natural that CWSs are more common in non-periphery areas than in periphery areas. In which distance ranges from the city center are CWSs located significantly more often? Does this tendency differ from other facilities?
  2. It is unclear whether there is a statistically significant difference in land use where CWSs are located between periphery and non-periphery areas. Is the trend different from facilities other than CWSs?
  3. There may be many restaurants and bus stops around not only CWSs but also other facilities. For example, you should make sure that they are more common around CWSs than randomly placed points.

L.375. “Here we found a dominance of CWS locations in ‘residential’ neighbourhood by 63 % in general, 62 % by non-peripheral and 69 % by peripheral character of the FAU.” Was there a statistically significant difference between non-peripheral and peripheral areas?

L.471-475. Any discussion not based on the results of this study should not be included here.

Minor comments

L.35, 151. This manuscript does not accurately describe the Garden City that aimed to bring work and housing closer together.

L.228. What does "pois free" mean here?

L.240. Are 6.096 and 50.326 average values?

L.259. The title of a figure should not start with "This map shows".

L.272,280. Check the figure number. The three pie charts (Figure 1-3) should be shown in a table for easy comparison.

L.293-302. These sentences should be moved to the Methods section.

Fig.4-7. The four bar charts should be shown in a table for easy comparison.

L.337. The title of a figure should not start with "This figure shows".

L.476-484. Bullet points should be changed to sentences.

Reviewer 2 Report

  • The abstract must be reformulated to provide more coherence to the research. The information provided is not clear and direct about the problem addressed, the methodology used and the results obtained, as well as the possible lines of future work on the subject and the solutions provided.
  • Section 1 (Introduction) provides information on the basis of the debate and theme addressed in the research. Due to its short length, it is recommended to merge paragraph 1 with paragraph 2.
  • In line 60-62 the concepts of workplace and place of living are introduced, but without entering to develop them. A further detailed and developed inclusion of both concepts is requested to provide a solid basis for the claims.
  • Line 63 states "Commute is reported to lead to unhappiness". Further development of data on this statement is requested, because by itself it does not provide solid information.
  • In line 94 the authors state "Instead, coworking could be an alternativeoption". This  statement  must  be justified. 
  • Line 101-102 reads "All abovementioned aspects are geographically related and the distance between place of residence, place of work – in a coworking space – and the provided services and offers matters". The geographical relationship of aspects mentioned must be justified, and linked with the concepts of residence, place and work.
  • Line 230 states "500 m  around  a coworking  space", and uses the buffer distanceof 500 meters for all research. A justification is requested on the use of that radius and not another radius in meters.
  • Paragraph 6 (limitations) may be merged with paragraph 7 (conclusion)

Reviewer 3 Report

The paper aims at investigating in more detail why and how coworking spaces are effectively constituted and what the effects of working in coworking spaces are. The novelty seems to be limited. At the same time, several aspects need more details and I would like to share with my specific comments and suggestions below:

1、The concept of coworking spaces should be introduced at the beginning of the paper.

2、A deep literature review should be given in the paper. When using open source data, data quality should be considered. e.g., 10.1111/j.1467-9671.2010.01203.x;

 10.1080/13658816.2020.1832228;  10.3390/ijgi2020507.

3、In line 230, the reason for choosing 500 meters should be briefly explained.

4、There are two Figure 1 in the paper. Please check the figure serial number carefully.

5、Figure 1 on page 7 should add map elements such as legend and scale. Figure 1 on page8, the figure label is not fully displayed.

6、In part of “Results”, the reasons for the results should be explained briefly

Reviewer 4 Report

I think that this is a good article. It contains the essential aspects to be published in a high impact scientific journal. It is a concise paper, with three clear objectives (research questions) that perfectly define the development, methodology and results.

I agree with the authors on the imitations of the paper: open source data, which may not have been validated throughout. Modelling assumptions that could be further detailed with topographic information or accessibility analysis with routing algorithms using a road network. The choice of POIs in peripheral and non-peripheral regions is questionable. But it is precisely the concreteness of these limits that gives value to the results. It also marks future lines of research

Reviewer 5 Report

This paper shows an interesting subject. The authors explain the methodology correctly, know how to use GIS and the explanation of results is correct. However, the paper needs to solve certain aspects:

1) It would be advisable to mention the area of study in the abstract section. This would help readers in order to locate the subject.

2) The theoretical background is correct, but it would be better if the authors deal with accessibility studies, in general.

3) It is necessary to justify why Germany is a good area in order to study CWS.

4) The map 1, (Figure 1), do not represent scale, therefore, it is not a map

5) Figure 1 is repeated. There are two Figure 1.

6) Figure 8, 9, 10 and 11. It would be advisable to introduce a little reference map in order to know where the areas are.

7) Limitations section. The comments are very brief. Please, the paper needs more debate related to previous articles with similar aspects. This is the main weak of the article. There is not a discussion section.

8) The conclusion has to be written in a discursive way, not answering questions